# Post-tetanic potentiation lowers the energy barrier for synaptic vesicle fusion independently of Synaptotagmin-1

Vincent Huson[1], Marieke Meijer[1], Rien Dekker[1], Mirelle ter Veer[2], Marvin Ruiter[2], Jan RT van Weering[1], Matthijs Verhage[1,2], Lennart Niels Cornelisse[1]*

[1]Department of Functional Genomics, Clinical Genetics, Center for Neurogenomics and Cognitive Research, Amsterdam University Medical Center- Location VUmc, Amsterdam, Netherlands; [2]Department of Functional Genomics, Center for Neurogenomics and Cognitive Research, VU University Amsterdam, Amsterdam, Netherlands

**Abstract** Previously, we showed that modulation of the energy barrier for synaptic vesicle fusion boosts release rates supralinearly (Schotten, 2015). Here we show that mouse hippocampal synapses employ this principle to trigger $Ca^{2+}$-dependent vesicle release and post-tetanic potentiation (PTP). We assess energy barrier changes by fitting release kinetics in response to hypertonic sucrose. Mimicking activation of the C2A domain of the $Ca^{2+}$-sensor Synaptotagmin-1 (Syt1), by adding a positive charge (Syt1[D232N]) or increasing its hydrophobicity (Syt1[4W]), lowers the energy barrier. Removing Syt1 or impairing its release inhibitory function (Syt1[9Pro]) increases spontaneous release without affecting the fusion barrier. Both phorbol esters and tetanic stimulation potentiate synaptic strength, and lower the energy barrier equally well in the presence and absence of Syt1. We propose a model where tetanic stimulation activates Syt1-independent mechanisms that lower the energy barrier and act additively with Syt1-dependent mechanisms to produce PTP by exerting multiplicative effects on release rates.

*For correspondence:
l.n.cornelisse@vu.nl

Competing interests: The authors declare that no competing interests exist.

## Introduction

Synaptic transmission is a highly dynamic process. Vesicle release rates change several orders of magnitude in response to $Ca^{2+}$ influx (*Lou et al., 2005*; *Schneggenburger and Neher, 2000*), and during repeated synaptic activity the number of vesicles released by an action potential (AP) rapidly change (*Abbott and Regehr, 2004*). Synaptic vesicle release is tightly controlled by specialized proteins, including SNAREs, SM proteins, and $Ca^{2+}$-sensors, among others (*Südhof, 2013*). Many of these are involved in the last step of the release process in which the fusion of the lipid membranes of the vesicle and presynaptic terminal occurs. This process can be described in terms of an energy landscape. Here a fusion energy barrier represents the activation energy that is required for the intermediate steps during membrane fusion, such as overcoming electrostatic repulsion of the membranes (*Ruiter et al., 2019*) and lipid stalk formation (*Dittman and Ryan, 2019*). In our previous work, we showed that additive changes in the fusion energy barrier produced supralinear changes in the vesicle fusion rate, as predicted by transition state theory (*Schotten et al., 2015*). We hypothesized that an energy barrier model could explain the supralinear relationship between $Ca^{2+}$-concentration and release rates (*Lou et al., 2005*) if $Ca^{2+}$ binding to the release sensor would decrease the fusion energy barrier (*Schotten et al., 2015*). The latter assumption has not been tested experimentally. Synaptotagmin-1 (Syt1) is the $Ca^{2+}$-sensor that is responsible for fast release in hippocampal synapses (*Geppert et al., 1994*; *Nishiki and Augustine, 2004a*; *Fernández-Chacón et al., 2001*). It was recently suggested that Syt1 lowers the fusion energy barrier by reducing electrostatic repulsion

between membranes upon binding of $Ca^{2+}$ (*Ruiter et al., 2019*). However, Syt1 is also involved in translocating vesicles to the plasma membrane (*de Wit et al., 2009*; *Kedar et al., 2015*; *Imig et al., 2014*; *Chang et al., 2018*), and inhibiting spontaneous and asynchronous release (*Kochubey and Schneggenburger, 2011*; *Nishiki and Augustine, 2004b*; *Littleton et al., 1994*; *Broadie et al., 1994*; *Maximov and Südhof, 2005*; *Huson et al., 2019*). Both these functions could potentially be involved in energy barrier modulation. Furthermore, in the absence of Syt1, slower $Ca^{2+}$-sensors that drive asynchronous release become prominent (*Geppert et al., 1994*; *Nishiki and Augustine, 2004a*; *Sun et al., 2007*). Synaptotagmin-7 has been shown to trigger asynchronous release (*Bacaj et al., 2013*; *Luo et al., 2015*; *Turecek and Regehr, 2018*), but is also identified as a $Ca^{2+}$ sensor for short-term facilitation (*Chen et al., 2017*; *Jackman et al., 2016*). The latter has been proposed to be due to its concerted action with Syt1 on the fusion energy barrier (*Schotten et al., 2015*; *Chen et al., 2017*; *Jackman and Regehr, 2017*).

In this study, we tested the assumptions that (1) Syt1 can reduce the fusion energy barrier when activated, and (2) PTP can be produced by activation of a second pathway that reduces the fusion barrier independently of Syt1, thereby amplifying the action of Syt1. Changes in the energy barrier were probed in several mutant variants of Syt1 and during PTP, using hypertonic sucrose (HS) stimulation. We found that mimicking activation of Syt1's $Ca^{2+}$-binding C2A domain reduced the energy barrier. Syt1's release inhibitory function acts independently from the energy barrier, indicating that additional release promoting factors contribute to spontaneous release. We found that after PTP or phorbol ester application the energy barrier was reduced, independently of the vesicle pool size and positional priming. Furthermore, this reduction did not require Syt1, and most likely is induced by activation of a second sensor. Altogether, these findings support a dual-sensor energy barrier model for supralinear $Ca^{2+}$-sensitivity of release and changes in synaptic strength after PTP.

## Results

### Synaptotagmin-1 inhibits spontaneous release without changing the fusion energy barrier

Syt1 is well established to be the fast $Ca^{2+}$-sensor for synaptic vesicle release in many synapses (*Geppert et al., 1994*; *Fernández-Chacón et al., 2001*). However, Syt1 is also known to act as an inhibitor of spontaneous and asynchronous release (*Kochubey and Schneggenburger, 2011*; *Nishiki and Augustine, 2004b*; *Littleton et al., 1994*; *Broadie et al., 1994*; *Maximov and Südhof, 2005*; *Huson et al., 2019*; *Xu et al., 2009*; *Bai et al., 2016*). This inhibition may act directly on the fusion machinery (*Ruiter et al., 2019*; *Chicka et al., 2008*; *Ramakrishnan et al., 2018*), suggesting an increase in the fusion energy barrier (*Ruiter et al., 2019*). Alternatively, Syt1 may inhibit a second high-affinity $Ca^{2+}$-sensor (*Kochubey and Schneggenburger, 2011*; *Xu et al., 2009*), reducing sensitivity to local $Ca^{2+}$-fluctuations (*Goswami et al., 2012*; *Ermolyuk et al., 2013*; *Emptage et al., 2001*), but likely not affecting the energy barrier. To discriminate between these two possibilities, we investigated whether the energy barrier in a resting synapse was altered in the absence of Syt1, comparing wild type (WT) and Syt1 KO glutamatergic hippocampal neurons. Syt1 deficiency did not affect the number of synapses (*Figure 1A,B*) or dendrite length (*Figure 1A,C*). However, electron microscopy (EM) revealed a significant reduction in membrane-proximal synaptic vesicles in Syt1 KO synapses (*Figure 1—figure supplement 1*), as observed before (*Imig et al., 2014*). Voltage clamp recordings revealed that spontaneous miniature excitatory post-synaptic current (mEPSC) frequency was more than doubled (*Figure 1E*), while first evoked EPSC charge was strongly reduced in Syt1 KO synapses, compared with WT (*Figure 1F,G*). As described previously, several synaptic parameters that provide information about priming and fusion (*Schotten et al., 2015*) can be obtained by fitting a minimal vesicle state model to HS responses. These include the pool of primed vesicles or readily releasable vesicle pool (RRP) as defined by the total number of vesicles released by an osmotic shock from 500 mM HS, maximal HS release rate ($k_{2,max}$), and change in the fusion energy barrier ($\Delta E_a$ (RT)) (For detailed methodology see: *Schotten et al., 2015*). In order to examine changes in the energy barrier and RRP, we applied a range of HS concentrations to WT and Syt1 KO synapses (*Figure 1H*). As shown previously (*Schotten et al., 2015*), kinetics of the responses became

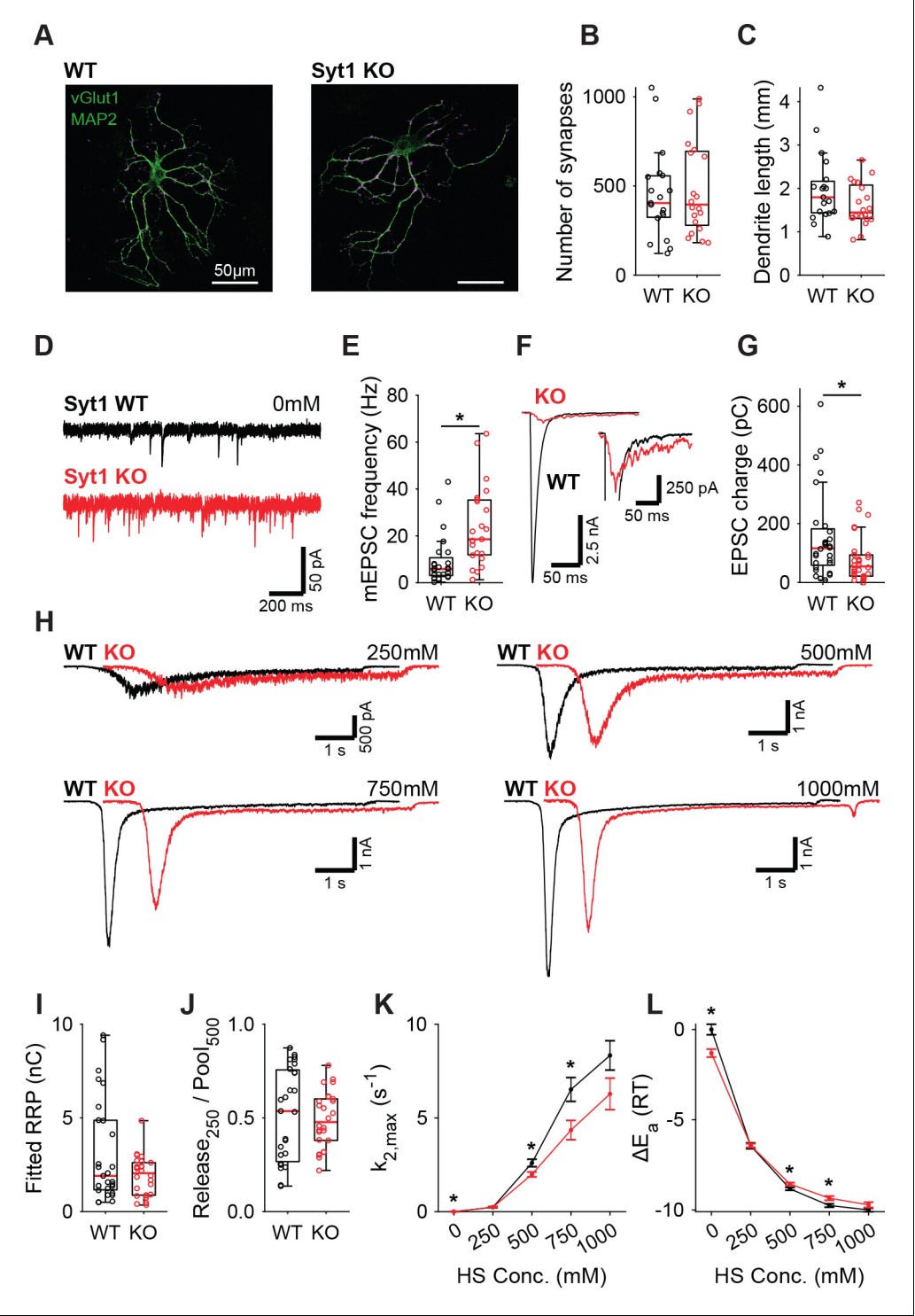

**Figure 1.** Increased spontaneous release in Syt1 KO does not correspond to a decrease in the fusion energy barrier. (**A**) Representative examples of WT and Syt1 KO neurons with vGlut (magenta) stained as a synapse marker and MAP2 (green) as a dendrite marker. (**B**) Boxplots of number of synapses, and (**C**) dendrite length per neuron in WT and Syt1 KO. (**D**) Representative traces of spontaneous release (0 mM HS), and (**E**) boxplot of spontaneous frequency in WT and Syt1 KO. (**F**) Representative traces of AP-evoked release in WT and Syt1 KO, overlaid, full view (left) and zoomed to the amplitude of Syt1 KO. (**G**) Boxplot of charge transferred during the first evoked EPSC. (**H**) Representative traces of HS-induced release in WT and Syt1 KO overlaid with 1 s offset, at 250 mM, 500 mM, 750 mM, and 1000 mM HS, and boxplots of (**I**) RRP charge estimated from 500 mM HS, (**J**) depleted

*Figure 1 continued on next page*

*Figure 1 continued*

RRP fraction at 250 mM HS in WT and Syt1 KO. (**K**) Plots (mean ± S.E.M.) of maximal HS release rates, and (**L**) change in the fusion energy barrier at different HS concentrations for WT and Syt1 KO. (*p<0.05, Wilcoxon rank sum test).

The online version of this article includes the following source data and figure supplement(s) for figure 1:

**Source data 1.** Statistics overview.
**Figure supplement 1.** Reduced number of docked vesicles in hippocampal Syt1 KO autaptic synapses.
**Figure supplement 1—source data 1.** Statistics overview.
**Figure supplement 2.** Additional HS parameters from WT and Syt1 KO neurons.
**Figure supplement 2—source data 1.** Statistics overview.

faster, while the delay of the onset of response decreased for increasing concentrations (*Figure 1—figure supplement 2*). The readily releasable pool (RRP), quantified from model fits of the current response to 500 mM HS, was not changed in Syt1 KO synapses (*Figure 1I*), despite the reduced number of membrane-proximal vesicles found with EM. The fraction of the RRP depleted by 250 mM sucrose (*Figure 1J*), a proxy for the energy barrier height (*Ruiter et al., 2019*; *Basu et al., 2007*), was not changed. Beyond 250 mM, HS-induced release rates even tended to be lower in Syt1 KO synapses than in WT (*Figure 1K*), corresponding to an increased fusion energy barrier under these conditions (*Figure 1L*). Therefore, we conclude that Syt1 in the non-activated state (i.e. without $Ca^{2+}$ bound) does not increase the energy barrier for synaptic vesicle fusion, despite its inhibitory effect on spontaneous release.

## 9Pro linker mutant confirms inhibition of spontaneous release by Synaptotagmin-1 is not through an increase in the energy barrier

Syt1 KO abolishes synchronous release triggering (*Geppert et al., 1994*; *Nishiki and Augustine, 2004a*; *Fernández-Chacón et al., 2001*) and impairs vesicle priming (*de Wit et al., 2009*; *Kedar et al., 2015*; *Imig et al., 2014*; *Chang et al., 2018*). To control for potential effects from these properties on HS-induced release, we used the Syt1 9Pro mutation to study the link between the inhibition of spontaneous release and the energy barrier in isolation. In this mutant, the flexible linker between Syt1's two $Ca^{2+}$-sensing C2 domains is fixed, selectively impairing inhibition of spontaneous release, without affecting evoked release (*Bai et al., 2016*; *Liu et al., 2014*). To further minimize activation of $Ca^{2+}$-sensors due to resting levels of intracellular $Ca^{2+}$, all recordings, with the exception of first evoked release, were done in 0 mM extracellular $Ca^{2+}$ with 20 µM BAPTA-AM. We expressed either Syt1 WT or Syt1 9Pro in Syt1 KO neurons, with Syt1 abundance in synapses exceeding endogenous levels to the same extent (*Figure 2—figure supplement 1*). We found a strong increase in mEPSC frequency in Syt1 9Pro synapses compared to Syt1 WT (*Figure 2A,B*). The first evoked EPSC was unaffected (*Figure 2C,D*). Applying 250 mM and 500 mM HS (*Figure 2E*, *Figure 2—figure supplement 2*), we found no difference in RRP (*Figure 2F*), or depleted RRP fraction at 250 mM (*Figure 2G*). Release rates in Syt1 9Pro synapses in the presence of BAPTA-AM remained higher at 0 mM sucrose (spontaneous release) compared with Syt1 WT (*Figure 2H*). However, at 250 mM and 500 mM HS we found no effect on the release rates and corresponding fusion energy barriers (*Figure 2H,I*). These data confirm that suppression of spontaneous release by Syt1 is not achieved by increasing the fusion energy barrier.

## Activation of Synaptotagmin-1's C2A domain lowers the fusion energy barrier

Next, we tested whether binding of $Ca^{2+}$ to Syt1's C2 domains lowers the energy barrier, as predicted by our energy barrier model for AP-evoked release (*Schotten et al., 2015*). To be able to test this with our HS assay, which is too slow to detect energy barrier changes during AP-induced $Ca^{2+}$ binding, we analyzed two different Syt1 mutants that mimic the effect of persistent activation by $Ca^{2+}$. We used the D232N mutation to neutralize a negatively charged residue in the C2A domain, which has been shown to increase $Ca^{2+}$-triggered release (*Xu et al., 2009*; *Pang et al., 2006*). Alternatively, we mimicked the $Ca^{2+}$-mediated association with phospholipids by increasing the hydrophobicity of the C2 domains, through insertion of tryptophan mutations in the C2 domains

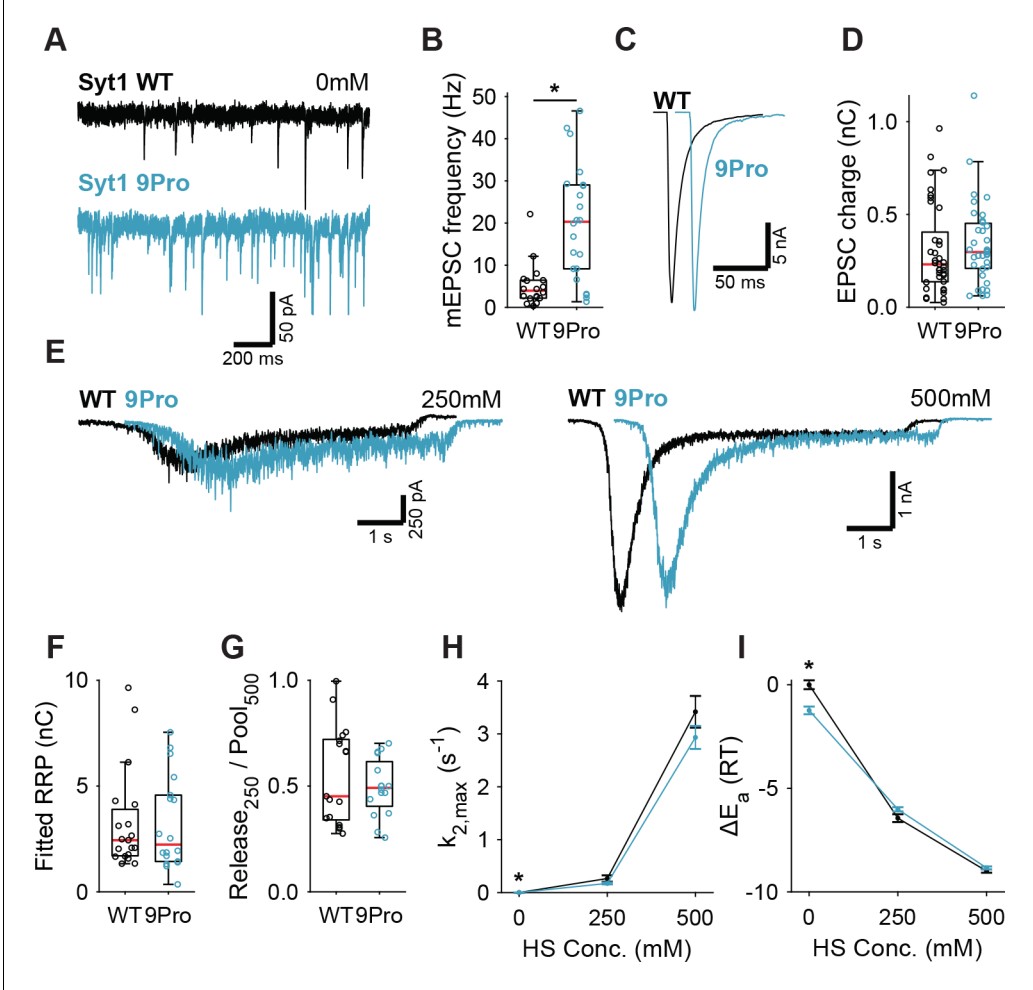

**Figure 2.** Syt1 9Pro mutation increases spontaneous release independent of the fusion energy barrier. (A) Representative traces of spontaneous release (0 mM HS), and (B) boxplot of spontaneous frequency in Syt1 WT and Syt1 9Pro expressing synapses. (C) Representative traces of AP-evoked release in Syt1 WT and Syt1 9Pro, overlaid with 20 ms offset, and (D) boxplot of charge transferred during the first evoked EPSC. (E) Representative traces of HS-induced release in Syt1 WT and Syt1 9Pro, overlaid with 1 s offset, at 250 mM and 500 mM HS, and boxplots of (F) RRP charge estimated from 500 mM HS, (G) depleted RRP fraction at 250 mM HS in Syt1 WT and Syt1 9Pro. (H) Plots (mean ± S.E.M.) of maximal HS release rates, and (I) change in the fusion energy barrier at different HS concentrations for Syt1 WT and Syt1 9Pro. (*$p < 0.05$, Wilcoxon rank sum test).

The online version of this article includes the following source data and figure supplement(s) for figure 2:

**Source data 1.** Statistics overview.
**Figure supplement 1.** Synaptic expression of Syt1 WT and mutant rescue constructs exceeds endogenous levels.
**Figure supplement 1—source data 1.** Statistics overview.
**Figure supplement 2.** Additional HS parameters from Syt1 WT and Syt1 9Pro expressing neurons.
**Figure supplement 2—source data 1.** Statistics overview.

(*Rhee et al., 2005*) (M173W, F234W, V304W, I367W; Syt1 4W). Given the high peak currents observed in previous preparations, we recorded the Syt1 D232N gain-of-function mutant, and matching Syt1 WT, in 2 mM extracellular $Ca^{2+}$ to avoid voltage-clamp artifacts. Synaptic expression of Syt1 WT or Syt1 D232N in Syt1 KO neurons exceeded endogenous Syt1 levels to the same extent (*Figure 2—figure supplement 1*). We confirmed previous findings in mass cultures (*Xu et al., 2009*; *Pang et al., 2006*) of increased mEPSC frequency (*Figure 3A,B*) and increased first evoked release in Syt1 D232N expressing synapses (*Figure 3C,D*). HS evoked release revealed no significant difference in RRP size (*Figure 3E,F*, *Figure 3—figure supplement 1A*), but the depleted RRP fraction at

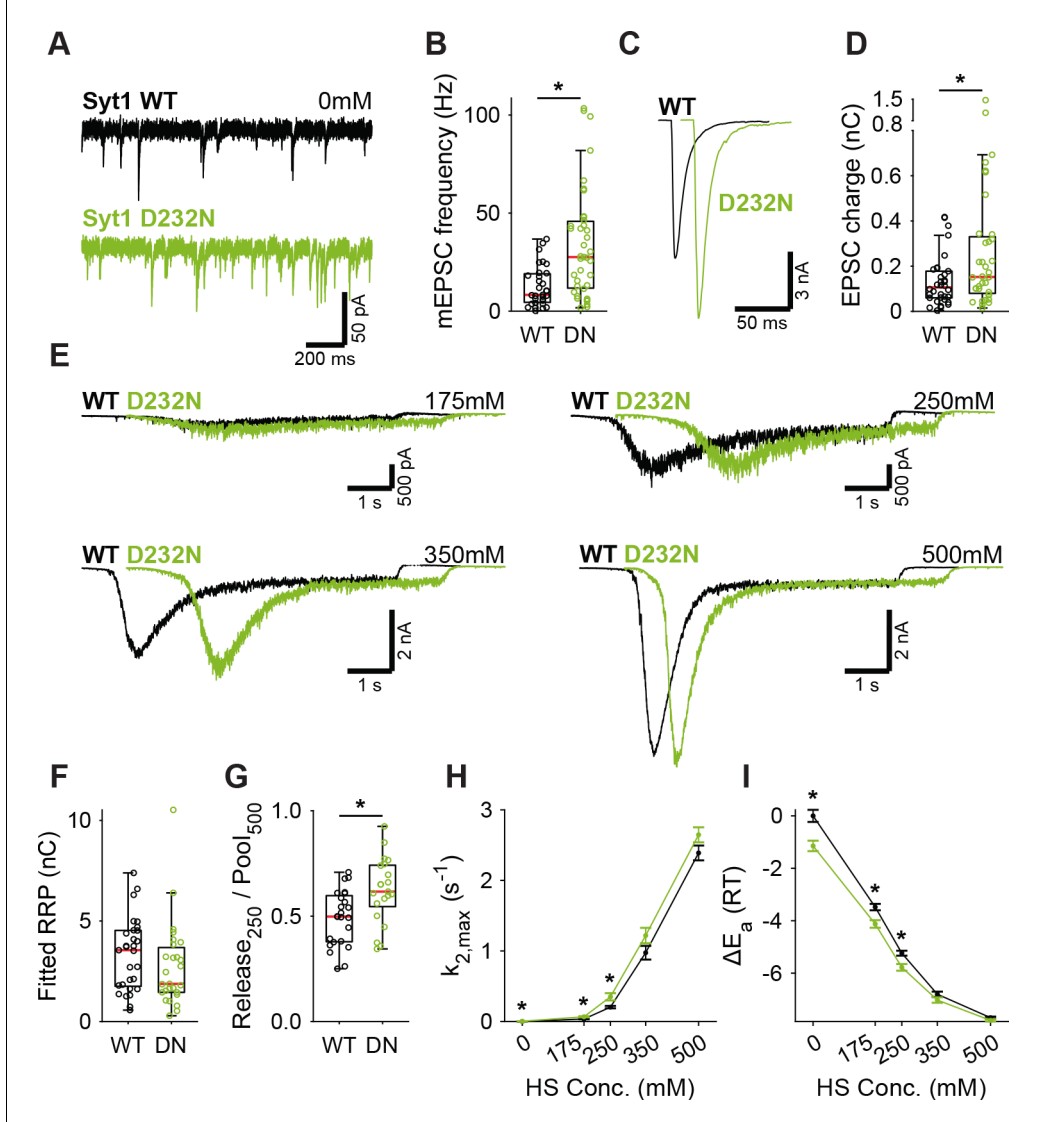

**Figure 3.** A decreased energy barrier increases vesicle fusion in Syt1 D232N expressing synapses. (**A**) Representative traces of spontaneous release (0 mM HS), and (**B**) boxplot of spontaneous frequency in Syt1 WT and Syt1 D232N expressing synapses. (**C**) Representative traces of AP-evoked release in Syt1 WT and Syt1 D232N, overlaid with 20 ms offset, and (**D**) boxplot of charge transferred during the first evoked EPSC. (**E**) Representative traces of HS-induced release in Syt1 WT and Syt1 D232N, overlaid with 1 s offset, at 175 mM, 250 mM, 350 mM, and 500 mM HS, and boxplots of (**F**) RRP charge estimated from 500 mM HS, (**G**) depleted RRP fraction at 250 mM HS in Syt1 WT and Syt1 D232N. (**H**) Plots (mean ± S.E.M.) of maximal HS release rates, and (**I**) change in the fusion energy barrier at different HS concentrations for Syt1 WT and Syt1 D232N. (*$p < 0.05$, Wilcoxon rank sum test). The online version of this article includes the following source data and figure supplement(s) for figure 3:

**Source data 1.** Statistics overview.
**Figure supplement 1.** Additional HS parameters from Syt1 WT and Syt1 D232N expressing neurons.
**Figure supplement 1—source data 1.** Statistics overview.
**Figure supplement 2.** Syt1 4W mutation increases HS release rates only at low concentration.
**Figure supplement 2—source data 1.** Statistics overview.
**Figure supplement 3.** Additional HS parameters from Syt1 WT and Syt1 4W expressing neurons.
**Figure supplement 3—source data 1.** Statistics overview.

250 mM was increased in Syt1 D232N expressing synapses (*Figure 3G*). Release rates at 175 mM and 250 mM HS were increased 1.7 to 1.9 fold, and the fusion energy barrier was reduced by 0.5 to 0.6 RT (*Figure 3H,I*), while a similar trend was observed for higher concentrations. Hence, constitutively activating Syt1 by removing negative charge from its $Ca^{2+}$-sensing domain increases vesicle fusion by decreasing the fusion energy barrier.

In Syt1 4W expressing synapses, we found a strong increase in mEPSC frequency (*Figure 3—figure supplement 2A,B*). No effect on first evoked charge was found (*Figure 3—figure supplement 2C,D*), but vesicular release probability was increased (*Figure 3—figure supplement 2E*). HS evoked release revealed no difference in RRP size (*Figure 3—figure supplement 2F,G*, *Figure 3—figure supplement 3A*). At 250 mM HS the depleted RRP fraction (*Figure 3—figure supplement 2H*) and the release rate (*Figure 3—figure supplement 2I*) were increased, both indicating a reduction of the fusion energy barrier by 0.5 RT (*Figure 3—figure supplement 2J*). However, we found no significant differences at other HS concentrations, and at 500 mM and 750 mM release rates trended towards a decrease (*Figure 3—figure supplement 2H*). Such a reversal of phenotype cannot readily be explained through energy barrier modulation alone, suggesting an interaction between HS and the 4W mutations (see discussion). Our data at 250 mM are in line with our findings with the Syt1 D232N mutant, corroborating the conclusion that activation of Syt1 C2A domain decreases the fusion energy barrier. Hence, supralinear $Ca^{2+}$-sensitivity through multiplicative effects on the release rate (*Lou et al., 2005*; *Schotten et al., 2015*) is likely supported by reductions of the energy barrier by Syt1.

## Induction of post-tetanic potentiation lowers the fusion energy barrier

Having established that activation of Syt1's C2A domain reduces the fusion energy barrier, we next investigated whether induction of short-term plasticity (STP) through repetitive stimulation also leads to a reduction. We assessed changes in the fusion barrier after induction of PTP in the presence and absence of Syt1. PTP is a form of STP which lasts tens of seconds to minutes (*Regehr, 2012*), allowing sufficient time to measure its effects using HS stimulation (*Stevens and Wesseling, 1999*; *Garcia-Perez and Wesseling, 2008*). It has previously been suggested that PTP acts by decreasing the fusion energy barrier (*Stevens and Wesseling, 1999*; *Garcia-Perez and Wesseling, 2008*). However, other studies have shown that an increase in the RRP after PTP can explain a large part of the potentiation of the EPSC (*Habets and Borst, 2005*; *Fioravante et al., 2011*). To resolve this, we induced synaptic release either through APs or HS before (Naive; *Figure 4A*), and 5 s after a train of 100 APs at 40 Hz (PTP; *Figure 4A*), to measure PTP, changes in the fusion energy barrier, and potentiation of RRP in the same cell. We have shown previously that our HS assay is sensitive enough to detect accelerated recovery of the RRP after 40 Hz stimulation (*He et al., 2017*). To maximize observable effects on the probability of vesicle release, we lowered extracellular $Ca^{2+}$ to 1 mM for this experiment. PTP increased EPSC charge by 39% (*Figure 4B,C*), even though the RRP, as assessed from 500 mM HS, was not fully recovered at this point (*Figure 4E,F*, *Figure 4—figure supplement 1A*). This resulted in a 41% increase in vesicle release probability, calculated as the ratio of the AP and HS-induced EPSC charges (*Figure 4D*). PTP increased the depleted RRP fraction at 250 mM (*Figure 4G*), and increased HS release rates 17% to 37% (*Figure 4H*), indicating about a 0.2–0.3 RT reduction of the fusion barrier (*Figure 4G,I*). These results confirm previous findings that PTP does not increase the RRP (*Stevens and Wesseling, 1999*; *Korogod et al., 2005*), and shows that it is associated with a decrease in the fusion energy barrier.

## PTP lowers the fusion energy barrier independently of Synaptotagmin-1

PTP induction is known to involve activation of Munc18-1 and PKC (*Fioravante et al., 2011*; *Wierda et al., 2007*), but the role of synaptotagmin in this pathway is less clear. We showed previously that phosphorylation by PKC plays a role in Syt1-, but not in Syt2 expressing synapses (*de Jong et al., 2016*). However, whether Syt1 is required for the energy barrier reduction after PTP is not known. To investigate this, we performed a similar set of PTP experiments in Syt1 KO autapses, using 4 mM extracellular $Ca^{2+}$. In Syt1 KO autapses, PTP also increased vesicle release probability (*Figure 5A–C*), while the RRP remained incompletely recovered (*Figure 5D,E*, *Figure 5—figure supplement 1A*). Additionally, similar to WT autapses, we observed after PTP an increase in

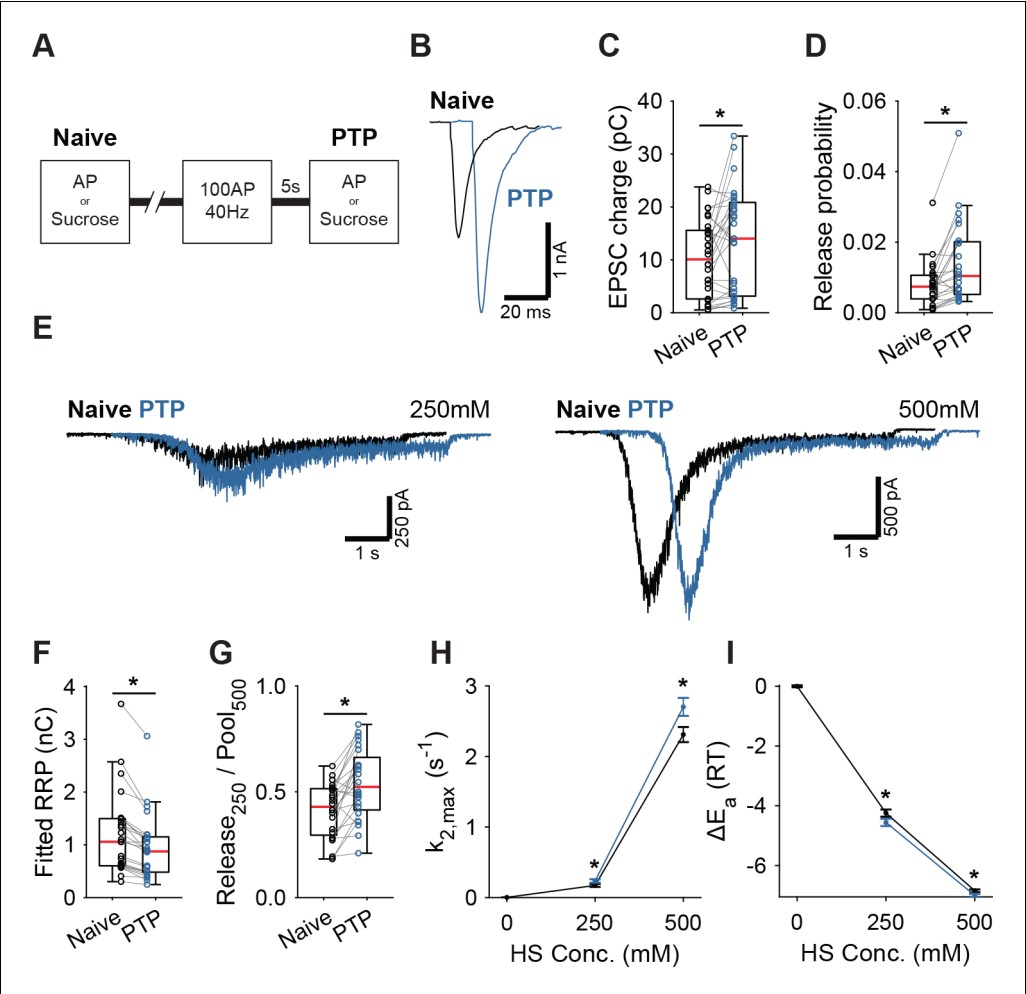

**Figure 4.** Post-tetanic potentiation causes a decrease in the fusion energy barrier. (**A**) Overview of protocol used to induce PTP. (**B**) Representative traces of AP-evoked release before (Naive) and after PTP in WT synapses, overlaid with 10 ms offset, and (**C**) boxplots of charge transferred during the first evoked EPSC and (**D**) release probability calculated by dividing the EPSC charge by the HS-derived RRP charge (**F**). (**E**) Representative traces of HS-induced release before and after PTP, overlaid with 1 s offset, at 250 mM and 500 mM HS, and boxplots of (**F**) RRP charge estimated from 500 mM HS, and (**G**) depleted RRP fraction at 250 mM HS before (Naive) and after PTP. (**H**) Plots (mean ± S.E.M.) of maximal HS release rates, and (**I**) change in the fusion energy barrier at different HS concentrations before (Naive) and after PTP. (*p<0.05, Wilcoxon signed-rank test).

The online version of this article includes the following source data and figure supplement(s) for figure 4:

**Source data 1.** Statistics overview.
**Figure supplement 1.** Additional HS parameters before and after PTP in WT neurons.
**Figure supplement 1—source data 1.** Statistics overview.

depleted RRP fraction at 250 mM (*Figure 5F*), and increased HS release rates (*Figure 5G*), corresponding to a decrease in the fusion energy barrier of 0.3–0.6RT (*Figure 5H*). Hence, reductions of the energy barrier due to PTP, do not require Syt1. Furthermore, as the RRP was not yet fully recovered after PTP, the contribution from the decreased energy barrier to potentiation of the EPSC is independent of priming. These findings suggest a model in which independent pathways for release cooperate to potentiate synaptic strength.

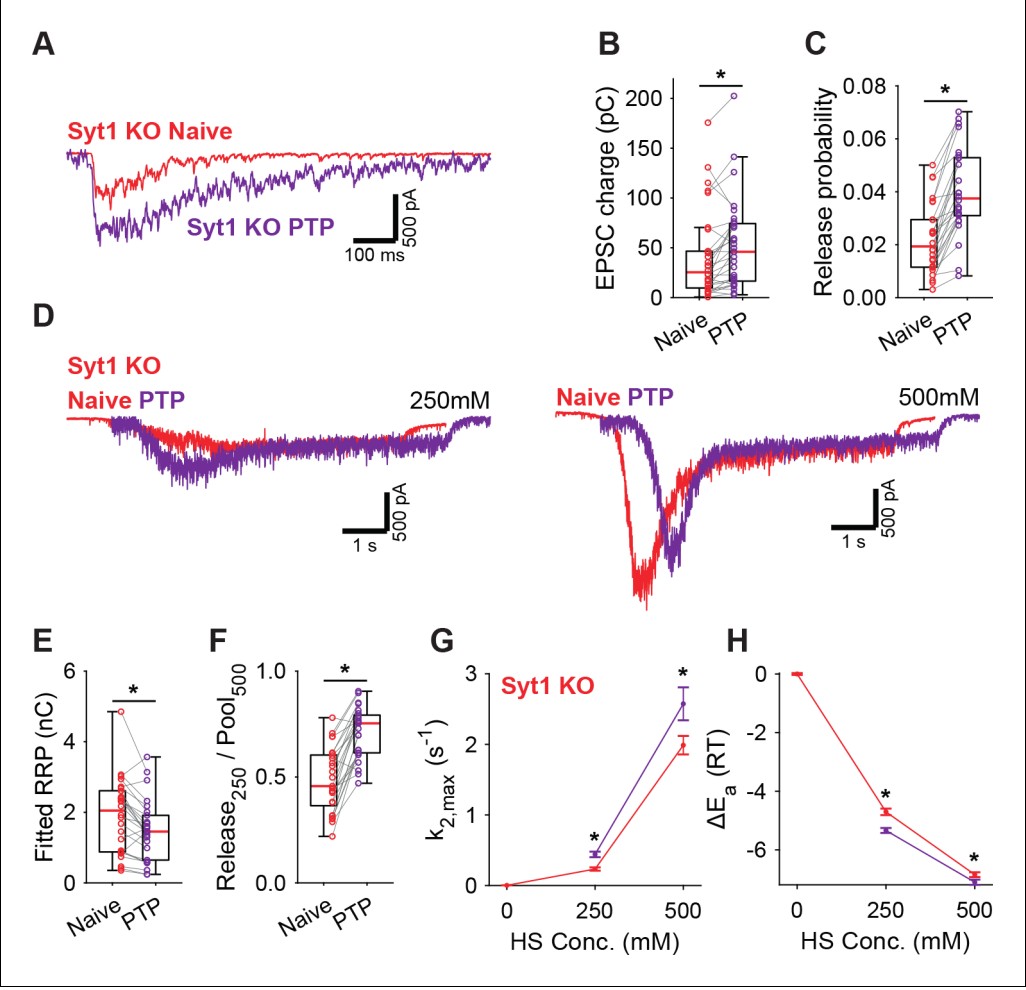

**Figure 5.** Post-tetanic potentiation decreases the fusion energy barrier independent of Syt1. (**A**) Representative traces of AP-evoked release before (Naive) and after PTP in Syt1 KO synapses, overlaid, and (**B**) boxplots of charge transferred during the first evoked EPSC and (**C**) release probability calculated by dividing the EPSC charge by the HS-derived RRP charge (**E**). (**D**) Representative traces of HS-induced release before and after PTP, overlaid with 1 s offset, at 250 mM and 500 mM HS, and boxplots of (**E**) RRP charge estimated from 500 mM HS, and (**F**) depleted RRP fraction at 250 mM HS before (Naive) and after PTP. (**G**) Plots (mean ± S.E.M.) of maximal HS release rates, and (**H**) change in the fusion energy barrier at different HS concentrations before (Naive) and after PTP. (*$p < 0.05$, Wilcoxon signed-rank test).

The online version of this article includes the following source data and figure supplement(s) for figure 5:

**Source data 1.** Statistics overview.

**Figure supplement 1.** Additional HS parameters before and after PTP in Syt1 KO neurons.

**Figure supplement 1—source data 1.** Statistics overview.

## Activation of the diacylglycerol pathway lowers the fusion energy barrier independently of Synaptotagmin-1

PTP acts via the same pathway as phorbol ester mediated potentiation (*Basu et al., 2007*; *Wierda et al., 2007*; *de Jong et al., 2016*; *Korogod et al., 2007*; *Wang et al., 2016*; *Rhee et al., 2002*). We showed previously that activation of the diacylglycerol (DAG) pathway with 1 µM phorbol 12,13-dibutyrate (PDBu) reduced the fusion energy barrier (*Schotten et al., 2015*). In another study, we showed that preventing Syt1 phosphorylation by PKC blocked potentiation of AP-induced release by phorbol esters but not the potentiation of HS responses (*de Jong et al., 2016*). To gather further proof for a Syt1-independent pathway for energy barrier reduction after PTP, we investigated whether the PDBu-induced reduction also occurred in the absence of Syt1. To this end, we

compared the effect of 1 µM PDBu on release in WT or Syt1 KO autapses. Spontaneous mEPSC frequency increased significantly in the presence of PDBu for both WT and Syt1 KO synapses (*Figure 6A–C*). PDBu application did not affect the RRP assessed with 500 mM HS (*Figure 6D,E*), and induced a similar increase in depleted RRP fraction at 250 mM, in WT and Syt1 KO synapses (*Figure 6F*). Release rates increased after application of PDBu (*Figure 6G,H*), associated with a similar decrease in the fusion energy barrier (*Figure 6I,J*) for both WT (0.4–0.6RT) and Syt1 KO (0.4–0.6RT). We conclude that both tetanic stimulation and activation of the DAG pathway reduces the fusion energy barrier, independently of Syt1. This supports a model for Ca$^{2+}$-dependent release and PTP where release sensors are activated independently, but produce multiplicative effects on the fusion rate through their additive effects on the fusion energy barrier.

## An energy barrier model for Ca$^{2+}$ induced vesicle release and post-tetanic potentiation

Previously, we proposed that transition state theory can be used to describe the process of vesicle fusion. In this framework, the fusion reaction is defined as the transition from the primed state to the fused state, with $E_a$ the activation energy that is required for the transition to occur, also referred to as the fusion energy barrier (*Figure 7A*; *Schotten et al., 2015*). According to the Arrhenius equation, there is an exponential relation between the activation energy and the rate of a reaction. This, implies that additive changes in the height of the fusion energy barrier lead to multiplicative effects on the fusion rate (*Schotten et al., 2015*). We showed that supralinear Ca$^{2+}$-dependence of release follows from this principle, when assuming an energy barrier reduction $E_f$ for each Ca ion that binds (*Figure 7C*, upper row) (*Lou et al., 2005*; *Schotten et al., 2015*). We now propose that the same framework can be used to describe asynchronous release and STP. This can be realized by adding additional release sensors to the model and adding their effects in the energy barrier domain (*Figure 7B*; *Schotten et al., 2015*; Equation 4). When multiple sensors are activated (*Figure 7A, B*), the new fusion energy barrier height $E_{new}$ is given by:

$$E_{new} = E_{old} - \sum_{i=1}^{N} E_i \tag{1}$$

with $N$ the number of sensors, $E_i$ the energy barrier reduction induced by sensor $i$, and $E_{old}$, the basal energy barrier height. Applying the Arrhenius Equation 7 gives the corresponding new fusion rate $k_{2,new}$:

$$\begin{aligned} k_{2,new} &= Ae^{-\frac{E_{old} - \sum_{i=1}^{N}\Delta E_i}{RT}} \\ &= Ae^{-\frac{E_{old}}{RT}}.e^{\frac{\sum_{i=1}^{N}\Delta E_i}{RT}} \\ &= l_+.f.g.h\dots \end{aligned} \tag{2}$$

with $l_+ = Ae^{-\frac{E_{old}}{RT}}$ the basal fusion rate, $f = e^{\frac{E_f}{RT}}$ the factor by which $l_+$ needs to be multiplied to account for activation of sensor $f$, and each additional sensor (i.e. $g,h,\dots$) after that (*Figure 7B*). $A$ is an empirical prefactor, $\bar{R}$ the gas constant, and $T$ the temperature. During synaptic activity these sensors may be directly activated by Ca$^{2+}$, or indirectly, through other pathways such as the diacylglycerol (DAG) pathway (*de Wit et al., 2009*). Through differences in activation and kinetic properties of different sensors, their combined effect could give rise to a diverse repertoire of vesicle release patterns in response to different patterns of presynaptic activity.

Based on our experimental findings, we propose a qualitative model for PTP, by combining the fast Ca$^{2+}$ sensor with five activation states from the allosteric model (*Lou et al., 2005*; *Schotten et al., 2015*), with a slow DAG sensor with one activation state. This gives a two-dimensional reaction scheme with in total 12 different vesicle states, each with its own associated fusion barrier (*Figure 7C*). As in the original allosteric model (*Lou et al., 2005*), substantial activation of the fast sensor only occurs by peak Ca$^{2+}$ levels reached during the AP due to its low Ca$^{2+}$-affinity, which synchronizes vesicle fusion to the moment of AP firing (*Figure 7C*; upper row). Activation of the slow sensor can occur by increased DAG levels triggered by elevated residual Ca$^{2+}$ after synaptic

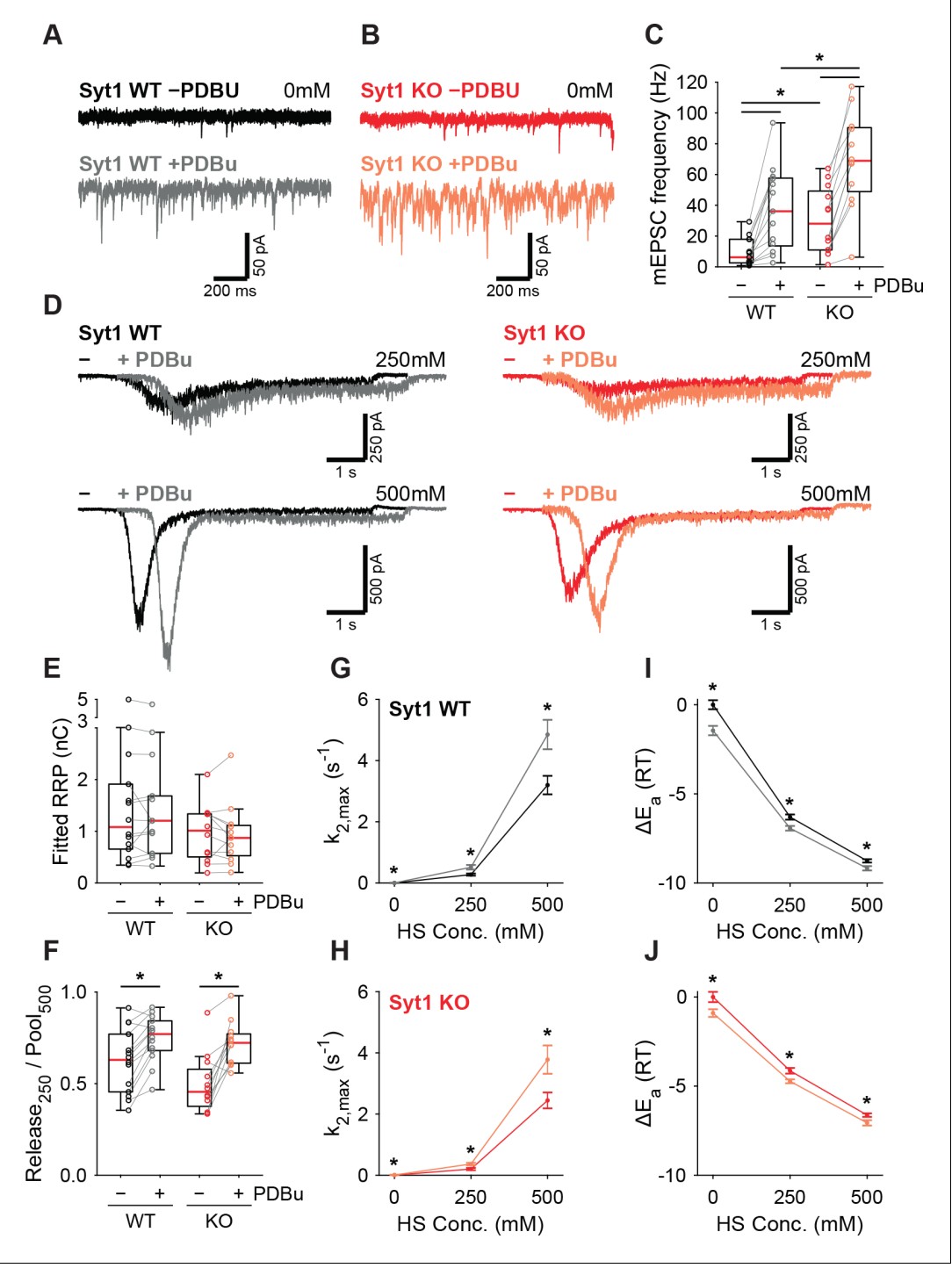

**Figure 6.** PDBu increases vesicles fusion by lowering the energy barrier in the presence or absence of Syt1. Representative traces of spontaneous release (0 mM HS) before (−PDBu; top) and after (+PDBu; bottom bath application of PDBu in WT (**A**) and Syt1 KO (**B**) synapses). (**C**) Boxplots of spontaneous release frequency in WT and Syt1 KO before and after PDBu. (**D**) Representative traces of HS-induced release at 250 mM and 500 mM HS, and boxplots of (**E**) RRP charge estimated from 500 mM HS, (**F**) depleted RRP fraction at 250 mM HS in WT and Syt1 KO before and after PDBu. (**G,H**) Plots (mean ± S.E.M.) of maximal HS release rates, and (**I,J**) change in the fusion energy barrier at different HS concentrations before and after PDBu, for WT (**G,I**) and Syt1 KO (**H,J**). (*$p < 0.05$, Wilcoxon rank sum test for independent and Wilcoxon signed-rank test for paired samples). The online version of this article includes the following source data and figure supplement(s) for figure 6:

*Figure 6 continued on next page*

*Figure 6 continued*

**Source data 1.** Statistics overview.
**Figure supplement 1.** Additional HS parameters before and after PDBu in Syt1 WT and KO neurons.
**Figure supplement 1—source data 1.** Statistics overview.

activity. The reduction of the fusion energy barrier $E_g$ after activation of the sensor multiplies the

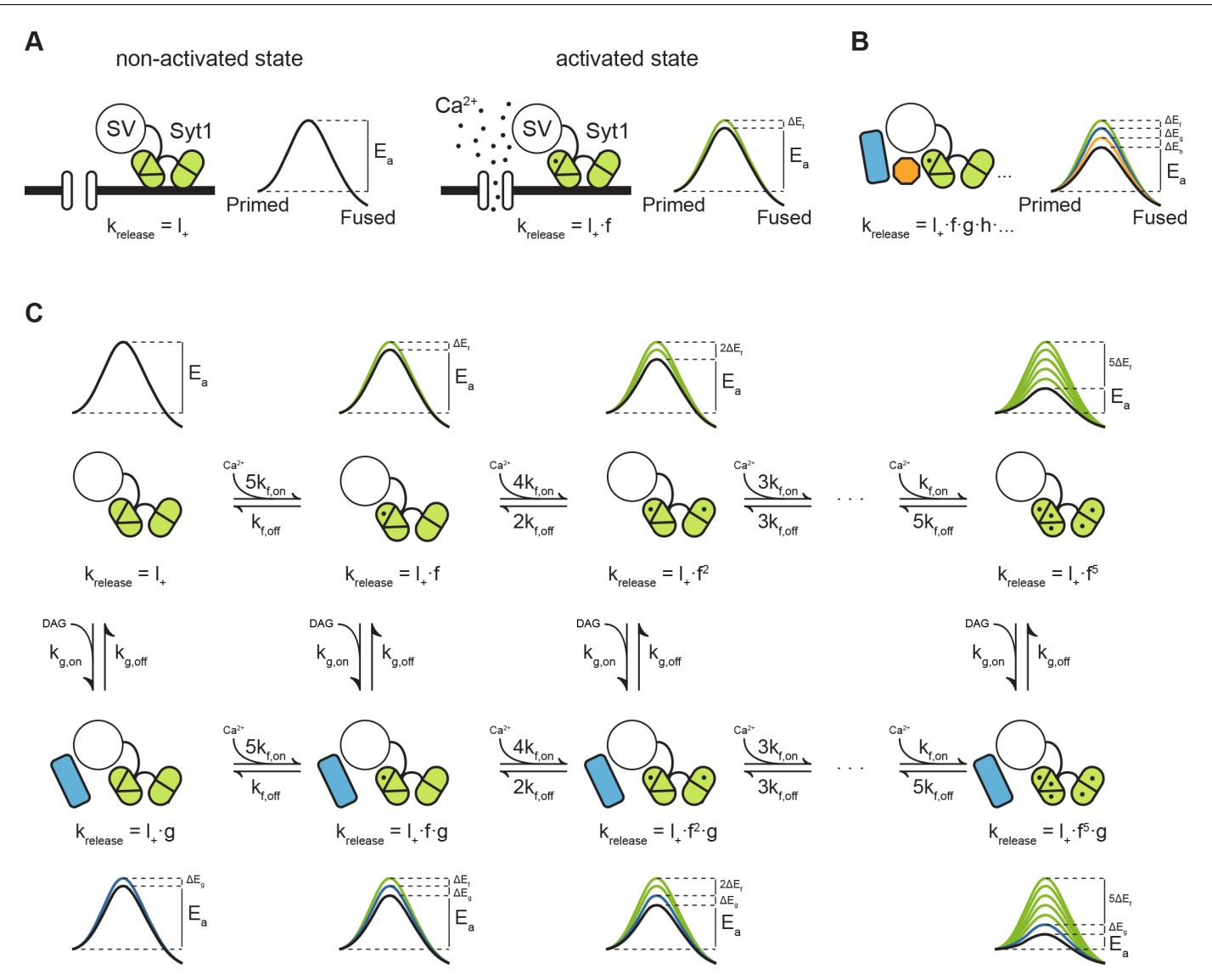

**Figure 7.** Modulation of the energy barrier through multiple sensors and activation states. (**A**) Fusion sensors in the non-activated state do not affect the fusion energy barrier ($E_a$), release rates ($k_{release}$) correspond to the energy barrier in the ground state ($l_+$; left). Upon activation by $Ca^{2+}$, Syt1 lowers the fusion energy barrier ($E_f$), multiplying release rates ($f$;right). (**B**) Multiple sensors in the activated state, each with separate additive effects on the energy barrier, provide multiplicative effects on release rates. (**C**) Binding of multiple $Ca^{2+}$ ions to Syt1 up to a maximum of 5, may be represented as multiple activation states additively lowering the energy barrier and multiplying release rates (top row). Additional activation of a second sensor ($g$) by DAG further expands the total number of states (bottom row), providing additional release pathways and increasing potential for plasticity.

The online version of this article includes the following source data and figure supplement(s) for figure 7:

**Figure supplement 1.** Syt1 D232N as a permanently activated sensor acting on the energy barrier.
**Figure supplement 1—source data 1.** Model parameter values.

spontaneous release rate by a factor $g = e^{\frac{Eg}{RT}}$ (Figure 1C; left column), leading to asynchronous release. PTP is induced when both sensors are activated simultaneously, for instance when several seconds after the train stimulation a new AP triggers the activation of the fast sensor, while the slow sensor is still activated by residual DAG. At this point fusion rates of the fast sensor are multiplied with the multiplication factor $g$ of the slow sensor (Figure 7C; bottom row). Such a model provides a simple explanation of how additional release promoters with small individual effects, may achieve meaningful changes in EPSC size (Chen et al., 2017; Jackman and Regehr, 2017; Schotten et al., 2015). We explored how the Syt1 D232N mutation could affect Ca²⁺ sensitivity in this model. Based on proposed electrostatic effects on the fusion barrier (Ruiter et al., 2019), the effect of adding one charge to the C2A domain can be modelled by increasing the basal fusion rate $l_+$ to $l_{+,new} = l_+ \ e^{\frac{0.5 \ E_f}{RT}} = l_+ \ f^{0.5}$. This is under the assumption that one charge is half as effective as two charges during Ca²⁺ binding (Ruiter et al., 2019), with $E_f$ the energy barrier reduction from one bound Ca²⁺ (Figure 7—figure supplement 1A,B). A two-fold increase in SNARE binding was also reported for this mutation, though only after Ca²⁺-binding (Pang et al., 2006). Therefore, this does not affect the basal release rate $l_+$, but can be modelled by replacing $f$ with $f_{D232N} = 2f$ (Figure 7—figure supplement 1A,B). Both parameter changes result in an increase in Ca²⁺-sensitivity of release, with higher release rates at all Ca²⁺ concentrations, and a similar small reduction in cooperativity (Figure 7—figure supplement 1B), while spontaneous release was only increased with a larger $l_+$.

## Discussion

Fusion rates change exponentially in response to linear changes of the fusion energy barrier. This makes modulation of the energy barrier a powerful principle for inducing quick and sizeable changes in synaptic strength. In this work, we propose that independent modulation of the energy barrier by different sensors in the synapse contributes to the supralinearity of Ca²⁺-dependent release and STP. In line with this idea, our results indicate that activation of the Ca²⁺-binding C2A domain of Syt1 potentiates release and decreases the energy barrier. Additionally, tetanic stimulation and PDBu both increase synaptic strength and decrease the fusion energy barrier, independent of Syt1. We propose that combined energy barrier reductions by Syt1 and the DAG-pathway contribute to the potentiation of EPSCs after PTP.

### Changes in spontaneous release do not directly correspond to changes in the energy barrier as assessed with hypertonic sucrose

When all sensors are in the non-activated state, the basal release rate constant $l_+$ in our model is associated with one effective energy barrier representing the fusion pathway. This includes intermediate steps such as stalk formation, hemifusion and pore formation (Figure 7A; Lou et al., 2005). In other studies its height has been estimated to be 30 $k_B$T based on experiments with pure lipid bilayers and from coarse-grained simulations of the underlying intermediate fusion states (François-Martin et al., 2017; Ryham et al., 2016; Smirnova et al., 2010). Previously, we concluded that changes in spontaneous release rates after genetic or biochemical manipulations did not correspond with the energy barrier shifts measured with HS (Schotten et al., 2015). Here we showed both in Syt1 KO (Figure 1) and Syt1 9Pro expressing synapses (Figure 2) that mEPSC frequencies were increased, in line with previous studies in autapses (Ruiter et al., 2019; Díez-Arazola et al., 2020; Huson and Cornelisse, 2019) and networks (Xu et al., 2009; Bai et al., 2016; Liu et al., 2014; Courtney et al., 2018). It remains enigmatic why others found a similar increase in spontaneous release in networks but not in autapses after Syt1 deletion (Liu et al., 2009; Wierda and Sørensen, 2014), but differences in culture protocol (Bekkers, 2020), or genetic background could play a role. Despite the increase in spontaneous release in our Syt1 KO and Syt1 9Pro expressing synapses, the fusion energy barrier assessed with HS was not changed (Figure 1L; Figure 2I). These findings indicate that for hippocampal autapses the rate constant for spontaneous release is not equal to the basal release rate constant $l_+$ as assessed with HS in the model, and that additional mechanisms may also contribute to AP independent release of vesicles. As spontaneous release is to a large extent Ca²⁺ dependent (Xu et al., 2009; Groffen et al., 2010), a possible mechanism may be rapid spontaneous Ca²⁺ fluctuations (Goswami et al., 2012; Ermolyuk et al., 2013; Emptage et al.,

*2001*). When such fluctuations occur locally at individual fusion sites, Ca²⁺ activation of a release sensor will reduce the energy barrier only locally and very briefly and not constitutively across all synapses simultaneously. We estimated previously that in such a scenario the frequency of Ca²⁺ fluctuations increases the release rate constant in an additive manner by 2-4 $10^{-4}$ s⁻¹ in WT autapses (*Schotten et al., 2015*). This will dominate the release rate at 0mM sucrose but is negligible compared to release rates induced by higher concentrations, corresponding to undetectable changes in the energy barrier. Interestingly, recording spontaneous release in 0mM extracellular Ca²⁺ and in the presence of the 20μM BAPTA-AM seemed to reduce its frequency less drastically as reported in other studies (compare *Figure 1E* and *2B*; *Xu et al., 2009*; *Courtney et al., 2018*). Moreover, differences in mEPSC frequency in Syt1 9Pro expressing synapses remained under these conditions. This might suggest that in our experiments the BAPTA loading was insufficient to block all spontaneous events. BAPTA-AM was applied after recording evoked release, with its incubation time restricted to 10 min to ensure the recording remained stable enough for HS measurements. Alternatively, part of the spontaneous release could be Ca²⁺ independent and through a different pathway than evoked release, either from a subset of synapses (*Peled et al., 2014*; *Melom et al., 2013*) or from a different vesicle pool (*Sara et al., 2005*; *Fredj and Burrone, 2009*). In this case, changes in the fusion barrier of this pathway could have been missed if spontaneous released vesicles are a small subset of the total pool, or when these vesicles are somehow less sensitive to HS stimulation.

## Syt1 does not inhibit spontaneous release by increasing the fusion energy barrier as assessed with hypertonic sucrose

Several mechanisms have been proposed to explain the inhibitory effect of Syt1 on spontaneous release (*Littleton et al., 1994*; *Broadie et al., 1994*; *Xu et al., 2009*; *Bai et al., 2016*). These include Syt1 clamping a second sensor for slow release (*Kochubey and Schneggenburger, 2011*; *Sun et al., 2007*), clamping fusion directly by arresting SNARE complexes (*Chicka et al., 2008*; *Ramakrishnan et al., 2018*), or increasing electrostatic repulsion between lipid membranes (*Ruiter et al., 2019*). The latter two mechanisms imply an increase in the energy barrier in the presence of Syt1, which we did not observe with our HS assay (*Figure 1*). Furthermore, Syt1 9Pro expressing synapses showed increased spontaneous release, but no changes in the energy barrier and normal AP-induced release. This indicates that Syt1's inhibitory role on spontaneous release is independent of its release-promoting role. We conclude that Syt1 does not increase the fusion energy barrier in its non-activated state. This conclusion does not support a model in which Syt1 suppresses spontaneous release by inhibiting the fusion step itself. Our data are most consistent with either a model where Syt1 clamps a slow sensor with high affinity for Ca²⁺, making the system less sensitive to spontaneous Ca²⁺ fluctuations, or with an additional fusion pathway for spontaneous release for a subset of vesicles, under the control of the clamping function of Syt1 (see discussion above).

## Activation of Syt1's Ca²⁺ binding domain decreases the fusion energy barrier

A central assumption of our model is that activation of Syt1 by Ca²⁺ binding lowers the fusion barrier (*Ruiter et al., 2019*; *Jackman and Regehr, 2017*; *Pérez-Lara et al., 2016*; *Park and Ryu, 2018*). Several competing models have been proposed for this (*Park and Ryu, 2018*). Both the Syt1 D232N mutant, where Ca²⁺-binding is mimicked, and the Syt1 4W mutant where hydrophobicity is increased, showed increased spontaneous and HS-induced release rates up to 250 mM HS, in line with a reduced fusion energy barrier. Interestingly, for 500 mM and 750 mM we also found a slightly reduced fusion barrier in WT cells compared to Syt1 KO cells (*Figure 1L*), possibly due to activation of Syt1 at basal Ca²⁺ levels. An electrostatic energy barrier model has been proposed, which assumes Ca²⁺-binding to Syt1 reduces the fusion barrier by diminishing electrostatic forces between opposing membranes (*Ruiter et al., 2019*). Adding a single positive charge to the C2A domain in the D232N mutant yielded a reduction between 0.5 and 0.6 RT. Assuming linear scaling of the fusion barrier with charge (*Ruiter et al., 2019*), this implies a 5 to 6 RT (or 2.9 to 3.5 kCal mol⁻¹) reduction when Syt1 is fully activated after binding 5 Ca²⁺ ions, adding 10 charges in total (*Lou et al., 2005*). This is in the same order of magnitude as the estimated 10 RT (5.9 kCal mol⁻¹) reduction during AP-induced release in hippocampal neurons (*Rhee et al., 2005*), but about three times smaller than the

17 RT reduction predicted for full occupation of Synaptotagmin 2 (Syt2) in the Calyx of Held (*Lou et al., 2005*). These results suggest that either differences exist in the efficiency of the fusion machinery in the two systems, or that our method has a limited ability to capture the full effect of Syt1 activation on the fusion barrier (see also discussion below). Alternatively, the energy barrier reduction and increased spontaneous release rate in the D232N mutant do not come from electrostatic effects, but from increased $Ca^{2+}$-dependent SNARE binding (*Pang et al., 2006*) at basal $Ca^{2+}$ levels. Modeling these scenarios (reduced electrostatic repulsion or increased $Ca^{2+}$-dependent SNARE binding) with the energy barrier model, predicted in both cases an increase in $Ca^{2+}$ sensitivity with little effect on cooperativity (*Figure 7—figure supplement 1*). These results are in line with previous studies in GABAergic neurons (*Xu et al., 2009*; *Pang et al., 2006*) with similar values for the apparent cooperativity. However, caution has to be taken to directly compare model predictions to these experimental studies, which show the IPSC amplitude (not peak release rate) as a function of the extracellular (not intracellular) $Ca^{2+}$ concentration. Interestingly, adding a positive charge to the C2A domain at a different position (D238N) was shown to reduce $Ca^{2+}$ sensitivity in the same study (*Pang et al., 2006*). This makes increased $Ca^{2+}$-dependent SNARE binding a more plausible explanation for the fusion barrier effects in the D232N mutant than reduced electrostatic repulsion, but more research is needed.

In contrast to our HS measurements after PDBu (*Schotten et al., 2015*) or PTP stimulation, we could not detect changes in the fusion barrier with concentrations beyond 250 mM for these Syt1 mutants. This may point to a possible interaction of the mechanisms by which sucrose and Syt1 induce vesicle fusion, which may be a limitation of the method for some specific conditions. Properties that contribute to the lateral pressure of the membrane, such as membrane fluidity, bilayer thickness, hydration state of lipid headgroups, and interfacial polarity and charge, can change in response to osmotic pressure (*Poolman et al., 2002*). This could render the membrane bending properties of Syt1 less effective or different at higher sucrose concentrations. In case of the Syt1 4W mutation, opposite effects of increased hydrophobicity of the C2 domains on membrane fusion could dominate at different osmotic conditions. Enhanced affinity for phospholipids may promote membrane-membrane interactions, increasing the chance to cross the fusion barrier during no or mild osmotic stress. At conditions where the energy barrier is substantially reduced, deeper insertions of the C2AB domains in the plasma membrane may lead to less curvature, rendering the vesicles less fusogenic (*Campelo et al., 2008*). Based on these considerations, we conclude that activation of Syt1 by $Ca^{2+}$ promotes vesicle release by reducing the fusion energy barrier.

## Energy barrier modulation contributes to PTP independently from vesicle priming

PTP and activation of the DAG-pathway are well established to increase synaptic strength (*Regehr, 2012*; *de Jong and Verhage, 2009*). Previous studies have suggested that energy barrier modulation contributes to STP (*Schotten et al., 2015*; *Chen et al., 2017*; *Jackman and Regehr, 2017*; *Basu et al., 2007*; *Stevens and Wesseling, 1999*; *Garcia-Perez and Wesseling, 2008*), while other studies suggested increased vesicle priming (*Habets and Borst, 2005*; *Fioravante et al., 2011*) or decreased un-priming (*Kobbersmed et al., 2020*). Using our HS assay, we found a decreased fusion barrier, but incomplete recovery of the RRP, after PTP (*Figures 4* and *5*). Furthermore, we found no change in RRP size after application of PDBu (*Figure 6E*), in line with previous studies (*Schotten et al., 2015*; *Basu et al., 2007*; *Wierda et al., 2007*; *Lou et al., 2008*), except one (*Stevens and Sullivan, 1998*). This indicates that in cultured hippocampal neurons lowering of the fusion barrier, and not increased vesicle priming or decreased un-priming, is a major factor in PTP. This is in line with the notion that activation of the DAG-pathway only increases the 'effective' pool size of AP releasable vesicles (*Lou et al., 2008*; *Lee et al., 2008*; *Lee et al., 2010*). This process, also referred to as 'superpriming', involves conversion of slowly releasing vesicles into rapidly releasing vesicles (*Lee et al., 2012*; *Lee et al., 2013*; *Taschenberger et al., 2016*; *Miki et al., 2016*; *Neher and Brose, 2018*), and could be interpreted as a transition of vesicles from a high- to low energy barrier state in the same RRP. PTP and PDBu stimulation reduced the fusion barrier about 0.2–0.6 RT and 0.4–0.6 RT, respectively (*Figures 4–6*). Although these effects are small compared to the estimated 30 RT fusion barrier for pure lipid bilayers (*François-Martin et al., 2017*) they correspond to a multiplication of the fusion rate by a factor 1.2–1.8 and 1.5–1.8 (*Equation 2*). These values are close to the 1.4 and 1.5–1.9 fold increase in evoked release after PTP (*Figure 4*) and

stimulation of the DAG pathway (*Basu et al., 2007*; *Wierda et al., 2007*; *Rhee et al., 2002*) in hippocampal autapses, and suggest a large contribution of fusion barrier modulation to tetanic-stimulation induced STP.

## PTP lowers the fusion energy barrier independently of Syt1

We postulated that PTP occurs through activation of a pathway that lowers the energy barrier independently of Syt1. Indeed, we found a similar reduction of the energy barrier after PDBu application or PTP induction in the presence or absence of Syt1. This reduction most likely requires activation of a second sensor (*Kochubey and Schneggenburger, 2011*; *Sun et al., 2007*). According to our model, a second sensor both amplifies the Syt1 induced fusion rates and triggers release by itself (*Figure 7*). However, the latter may occur at a slower rate than Syt1 induced release, and represent the increase of asynchronous release after tetanic stimulation (*Maximov and Südhof, 2005*; *Huson et al., 2019*) and the increase in spontaneous release after PDBu (*Figure 6*). For PTP different sensors and/or pathways may be involved (*Regehr, 2012*; *de Jong and Verhage, 2009*), including the DAG pathway. DAG-analogs enhance spontaneous release, AP-induced release, and vesicle priming and reduce the fusion energy barrier (*Lou et al., 2005*; *Schotten et al., 2015*; *Basu et al., 2007*; *Habets and Borst, 2005*; *Fioravante et al., 2011*; *Wierda et al., 2007*; *de Jong et al., 2016*; *Korogod et al., 2007*; *Lou et al., 2008*; *Hori et al., 1999*). Munc13 is directly activated by DAG (*Betz et al., 1998*), contributes to short-term synaptic plasticity (*Basu et al., 2007*; *Rhee et al., 2002*; *Shin et al., 2010*; *Lipstein et al., 2012*; *Junge et al., 2004*; *Rosenmund et al., 2002*) and modulates the fusion barrier (*Basu et al., 2007*). PKCs have been identified as relevant DAG targets in hippocampal neurons (*Wierda et al., 2007*) and the Calyx of Held (*Fioravante et al., 2011*). Phosphorylation of synaptotagmin by PKC has been shown to play a role in STP in Syt1-, but not Syt2-, expressing synapses (*de Jong et al., 2016*). Interestingly, preventing phosphorylation of Syt1 did not affect the fusion barrier and an effect on priming was proposed.

In conclusion, lowering of the fusion energy barrier during PTP occurs independently of Syt1, likely by $Ca^{2+}$ and/or DAG activation of an additional, slow sensor. However, an alternative mechanism is a reduction of electrostatic repulsion between opposing negative membranes through accumulation of residual $Ca^{2+}$ (*Ruiter et al., 2019*; *Jeremic et al., 2004*).

## Fusion energy barrier model for PTP

Several mechanisms have been proposed for synaptic plasticity, including ways to increase the presynaptic $Ca^{2+}$ signal (for review see *Jackman and Regehr, 2017*), (activity dependent) channel-attachment of vesicles (*Pan and Zucker, 2009*; *Böhme et al., 2018*), $Ca^{2+}$ dependent vesicle priming (*Habets and Borst, 2005*; *Fioravante et al., 2011*) or inhibition of un-priming (*Kobbersmed et al., 2020*), and tightening of the SNARE-complex (*Neher and Brose, 2018*). Our model employs a single mechanistic principle, multiplicative modulation of fusion rates through additive modulation of the energy barrier, to describe supralinear $Ca^{2+}$-sensitivity of release and PTP. A different dual sensor model was proposed before in which release can only occur from the vesicle states where one or both of the sensors are fully activated by $Ca^{2+}$ (*Sun et al., 2007*). This model can accurately describe $Ca^{2+}$-sensitivity of release in Syt1 WT and KO synapses but not STP. In our model, release can occur from all states, also when sensors are partly activated. This provides a general framework for STP in which multiple sensors can cooperate in controlling release through their additive effect on the fusion barrier, without requiring direct interaction. Within this framework, neurons can implement different forms of STP within the same synapses, by expressing different combinations of sensors, all with their own dynamic properties (*de Jong and Verhage, 2009*). In addition, it can reconcile the dual role of slow sensors in release, being both a sensor for asynchronous release and STP. For paired-pulse facilitation, a form of STP occurring at a shorter time scale than PTP (*de Jong and Verhage, 2009*), such a dual role has been suggested for Syt7 in corticothalamic and hippocampal synapses (*Bacaj et al., 2013*; *Luo et al., 2015*; *Turecek and Regehr, 2018*; *Chen et al., 2017*; *Jackman et al., 2016*; *Jackman and Regehr, 2017*; *Luo and Südhof, 2017*). Its role as a facilitation sensor, with a multiplicative effect on the fusion rate, is supported by the finding that multiplication of membrane bound Syt1 and Syt7, as a proxy for the activation of both sensors, can account for the observed facilitation (*Jackman and Regehr, 2017*). However, a dual sensor model could not explain all features of vesicle release in the *Drosophila* neuromuscular junction

(*Kobbersmed et al., 2020*). In its current form our model for PTP is a qualitative and reduced model which does not include ($Ca^{2+}$-dependent) priming of vesicles, spatial distribution of vesicles or detailed $Ca^{2+}$ dynamics, nor the effect of Syt1 or other sensors on these processes. Furthermore, it does not explicitly describe all successive steps in the fusion pathway (*Dittman and Ryan, 2019*) and the different roles of Syt1 during these steps (*Chang et al., 2018*; *Zhou et al., 2015*; *Zhou et al., 2017*), but models fusion as a single step using one effective energy barrier. Despite these limitations, the model is able to explain the increase in fusion rate upon $Ca^{2+}$ binding and increase in release probability during PTP, the latter with the relatively small changes in fusion barrier we found. This suggests that, while other mechanisms also may contribute, modulation of the fusion barrier is an important mechanism by which a synapse controls its efficacy over a short time scale.

# Materials and methods

**Key resources table**

| Reagent type (species) or resource | Designation | Source or reference | Identifiers | Additional information |
|---|---|---|---|---|
| Strain, strain background (*M. musculus*) | C57BL/6 | Charles River Laboratories | RRID:IMSR_CRL:27 | |
| Strain, strain background (*M. musculus*) | Synaptotagmin-1 KO | Geppert M, Goda Y, Hammer RE, Li C, Rosahl TW, Stevens CF, Südhof TC: Synaptotagmin I: a major Ca2+ sensor for transmitter release at a central synapse. *Cell* 1994, 79:717–27. | DOI: 10.1016/0092-8674(94)90556-8 PMID:7954835 | |
| Transfected construct (*M. musculus*) | pSyn-Syt1-WT-2A-EGFP | Huson V, Boven MA van, Stuefer A, Verhage M, Cornelisse LN, van Boven MA, Stuefer A, Verhage M, Cornelisse LN, Boven MA van, et al.: Synaptotagmin-1 enables frequency coding by suppressing asynchronous release in a temperature dependent manner. *Sci Rep* 2019, 9:11341. | DOI: 10.1038/s41598-019-47487-9 PMID:31383906 | |
| Transfected construct (*M. musculus*) | pLOXSyn-Syt1-9Pro-Syn-GFP | Liu H, Bai H, Xue R, Takahashi H, Edwardson JM, Chapman ER: Linker mutations reveal the complexity of synaptotagmin one action during synaptic transmission. *Nat Neurosci* 2014, 17:670–7. | DOI:10.1038/nn.3681 PMID:24657966 | |
| Transfected construct (*M. musculus*) | pSyn-Syt1(D232N)-IRES-EGFP | This paper | | Generated in-house with Quickchange (Stratagene). Construct available upon request. |
| Transfected construct (*M. musculus*) | pSyn-Syt1 (M173W,F234W, V304W,I367W)-IRES-EGFP | This paper | | Generated in-house with Quickchange (Stratagene). Construct available upon request. |
| Antibody | anti-MAP2 polyclonal chicken | Abcam | Cat#: ab5392 RRID:AB_2138153 | (1:10 000) |
| Antibody | anti-VGLUT1 polyclonal rabbit | Synaptic Systems | Cat#: 135 302 RRID:AB_887877 | (1:500) |
| Antibody | anti-VGLUT1 polyclonal guinea pig | Millipore | Cat#: AB5905 RRID:AB_2301751 | (1:5000) |
| Antibody | anti-VAMP2 monoclonal mouse | Synaptic Systems | Cat#: 104 211 RRID:AB_2782975 | (1:1000) |

*Continued on next page*

*Continued*

| Reagent type (species) or resource | Designation | Source or reference | Identifiers | Additional information |
|---|---|---|---|---|
| Antibody | anti-Synaptotagmin-1 polyclonal rabbit | T.C. Südhof, Stanford, CA | W855 | (1:1000) |
| Antibody | anti-GFP polyclonal chicken | AVES | Cat#: 1020 RRID:AB_10000240 | (1:2000) |
| Chemical compound, drug | BAPTA-AM | Sigma Aldrich | Cat#: A1076 | |
| Chemical compound, drug | Phorbol-12,13-dibutyrate (Calbiochem) | VWR | Cat#: 80055–388 | |
| Software, algorithm | MATLAB | Mathworks | RRID:SCR_001622 | Version R2020a |
| Software, algorithm | ImageJ/Fiji | Fiji (http://fiji.sc/) | RRID:SCR_002285 | Version 1.52 f |
| Software, algorithm | pClamp | Molecular Devices | RRID:SCR_011323 | Version 10.3 |

## Animals

Neuronal cultures were prepared from embryonic day 18 (E18) pups of both sexes, obtained by caesarean section of pregnant female mice. For this, previously described Synaptotagmin-1 knockout (*Geppert et al., 1994*) or C57BL/6 mouse lines were used. Newborn pups (P0-P1) from Winstar rats were used for glia preparations. Animals were housed and bred according to institutional and Dutch governmental guidelines, and all procedures are approved by the ethical committee of the Vrije Universiteit, Amsterdam, The Netherlands.

## Dissociated neuronal cultures and lentiviral infection

Hippocampi from WT and Syt1 KO mice were isolated, collected in ice-cold Hank's buffered salt solution (HBSS; Sigma) buffered with 1 mM HEPES (Invitrogen), and digested for 20 min with 0.25% trypsin (Invitrogen) at 37˚C. After washing, neurons were dissociated using a fire-polished Pasteur pipette and resuspended in Neurobasal medium supplemented with 2% B-27, 1% HEPES, 0.25% GlutaMAX, and 0.1% Penicillin-Streptomycin (all Invitrogen). Neurons were counted in a Fuchs-Rosenthal chamber and plated at 1.5K per well in a 12-well plate. Neuronal cultures were maintained in Neurobasal medium supplemented with 2% B-27, 1% HEPES, 0.25% GlutaMAX, and 0.1% Penicillin-Streptomycin (all Invitrogen), at 37˚C in a 5% CO2 humidified incubator.

Autaptic hippocampal cultures were prepared as described previously (*Meijer et al., 2018*). Briefly, micro-islands were prepared with a solution containing 0.1 mg/ml poly-D-lysine (sigma), 0.7 mg/ml rat tail collagen (BD Biosciences) and 10 mM acetic acid (Sigma) applied with a custom-made rubber stamp (dot diameter 250 μm). Next, rat astrocytes were plated at 6–8K per well in prewarmed DMEM (Invitrogen), supplemented with 10% FCS, 1% Penicillin-Streptomycin and 1% nonessential amino acids (All Gibco).

For rescue experiments, Syt1 KO neurons were infected at DIV4 with a synapsin-promoter-driven lentiviral vector expressing either Syt1 9Pro (residues 264–272 replaced with nine proline residues; kindly provided by dr. Edwin Chapman, Howard Hughes Medical Institute, Madison, WI, USA), Syt1 D232N, Syt1 4W (M173W, F234W, V304W, I367W), or wild type Syt1. The experimental groups were masked during the experiment. The code was broken after statistical analysis.

## Electrophysiology

Whole-cell voltage-clamp recordings ($V_m = -70$ mV) were performed at room temperature with borosilicate glass pipettes (2–5 MΩ) filled with (in mM) 125 $K^+$-gluconic acid, 10 NaCl, 4.6 MgCl$_2$,4 K2-ATP, 15 creatine phosphate, 1 EGTA, and 10 units/mL phosphocreatine kinase (pH 7.30). External solution contained in mM: 10 HEPES, 10 glucose, 140 NaCl, 2.4 KCl, 4 MgCl$_2$ (pH = 7.30, 300 mOsmol). 4 mM CaCl$_2$ was used externally in all experiments, unless otherwise specified. Inhibitory neurons were identified and excluded based on the decay of postsynaptic currents. Recordings were acquired with a MultiClamp 700B amplifier, Digidata 1440 A, and pCLAMP 10.3 software (Molecular

Devices). Only cells with an access resistance <15 MΩ (80% compensated) and leak current of <300 pA were included. EPSCs were elicited by a 0.5 ms depolarization to 30 mV.

Hypertonic sucrose stimulation was performed as described previously (Schotten et al., 2015). Briefly, gravity infused external solution was alternated with 7 s of perfusion with hypertonic solution by rapidly switching between barrels within a custom-made tubing system (FSS standard polyamine coated fused silica capillary tubing, ID 430 µm, OD550 µm) attached to a perfusion Fast-Step delivery system (SF-77B, Warner instruments corporation) and directed at the neuron. Solution flow was controlled with an Exadrop precision flow rate regulator (B Braun). Multiple sucrose solutions with various concentrations were applied to the same cell, taking a 1–2 min rest period in between solutions to accommodate complete recovery of RRP size. In between protocols, a constant flow of external solution was applied to the cells. Multiple sucrose solutions with various concentrations were applied to the same cell, taking a 1–2 min rest period,>3 min in case of post-tetanic potentiation protocols. The order of sucrose solutions was alternated between neurons to avoid systematic errors due to possible rundown of RRP size after multiple applications.

For experiments including the cell permeable $Ca^{2+}$ chelator BAPTA-AM, after recording of the first evoked response, cells were incubated for 10 min with 20 µM BAPTA-AM (Sigma) in bath and external solution was exchanged for 0 mM $CaCl_2$. A decrease in spontaneous release during incubation was used as a positive control. For PDBu experiments, sucrose applications were performed as usual, after which neurons were incubated with 1 µM PDBu (Calbiochem), and sucrose applications were repeated.

## Immunocytochemistry

Hippocampal neurons were fixed with 3.7% formaldehyde (Electron Microscopy Sciences) after two weeks in culture. After washing with PBS, cells were permeated with 0.5% Triton X-100 for 5 min and incubated in 2% normal goat serum/0.1% Triton X-100 for 30 min to block aspecific binding. Cells were incubated for 1 hr at room temperature with primary antibodies directed against MAP2 and vGlut1 to visualize dendrite morphology and synapses. The following antibodies were used: polyclonal chicken anti-MAP2 (1:10 000, Abcam), polyclonal rabbit vGlut1 (1:500, SySy), polyclonal guinea pig vGlut1 (1:5000, Millipore), monoclonal mouse VAMP2 (1:1000, SySy), polyclonal rabbit Synaptotagmin-1 (1:1000; W855; a gift from T. C. Südhof, Stanford, CA), or polyclonal chicken GFP (1:2000, AVES). After washing with PBS, cells were incubated for 1 hr at room temperature with second antibodies conjugated to Alexa dyes (1:1000, Molecular Probes) and washed again. Coverslips were mounted with DABCO-Mowiol (Invitrogen) and imaged with a confocal LSM510 microscope (Carl Zeiss) using a 40 × oil immersion objective with 0.7 × zoom at 1024 × 1024 pixels. Neuronal morphology was analyzed using a published automated image analysis routine (Schmitz et al., 2011), and ImageJ.

## Electron microscopy

Autaptic hippocampal neuron cultures of WT and Syt-1 KO mice (E18) grown on glass cover slips were fixed (DIV14) for 90 min at room temperature with 2.5% glutaraldehyde in 0.1 M cacodylate buffer (pH 7.4). After fixation, cells were washed three times for 5 min with 0.1 M cacodylate buffer (pH 7.4), post-fixed for 1 hr at room temperature with 1% OsO4/1% KRu(CN)6. After dehydration through a series of increasing ethanol concentrations, cells were embedded in Epon and polymerized for 48 hr at 60℃. After polymerization of the Epon, the coverslip was removed by alternately dipping it in hot water and liquid nitrogen. Cells of interest were selected by observing the flat Epon-embedded cell monolayer under the light microscope and mounted on pre-polymerized Epon blocks for thin sectioning. Ultrathin sections (80 nm) were cut parallel to the cell monolayer, collected on single-slot, formvar-coated copper grids, and stained in uranyl acetate and lead citrate using a LEICA EM AC20 stainer. Synapses were randomly selected at low magnification using a JEOL 1010 electron microscope. For each condition, the number of docked synaptic vesicles, total synaptic vesicle number, postsynaptic density and active zone length were measured on digital images taken at 80,000-fold magnification using analySIS software (Soft Imaging System). The observer was blinded for the genotype. For all morphological analyses, we selected only clearly recognizable synapses with intact synaptic plasma membranes with a recognizable pre- and postsynaptic area and clearly defined synaptic vesicle membranes. Synaptic vesicles were defined as docked if

there was no distance visible between the synaptic vesicle membrane and the active zone membrane. The active zone membrane was recognized as a specialized part of the presynaptic plasma membrane that contained a clear density opposed to the postsynaptic density and docked synaptic vesicles. Cells were cultured from six different WT and seven different Syt-1 KO mice. Approximately 25 synapses were analyzed per culture stemming from one animal.

## Data analysis

Offline analysis was performed using custom-written software routines in Matlab R2018b (Mathworks). Software routines for analysis of mEPSCs and electrically evoked release is available at https://github.com/vhuson/viewEPSC (*Huson and Cornelisse, 2019*), software for analysis of HS evoked release has been made available previously (*Schotten et al., 2015*; https://doi.org/10.7554/eLife.05531.031). In all figures, stimulation artefacts have been removed. For evoked release, total charge was calculated by integrating the current from the end of the stimulation until the start of the next pulse. HS-induced responses were fitted with a minimal vesicle state model as described previously (*Schotten et al., 2015*). This method corrects dynamically for ongoing priming opposed to the use of fixed priming rates in other methods. Furthermore, it provided direct estimates of fusion and priming rates, RRP size, and changes in energy barrier. Parameters describing the kinetics of HS responses were given in supplementary figures for completeness. HS integral was obtained by integrating the full 7s fitted trace. Rise time and time-to-peak were calculated using the minimum of the fitted trace. The delay of HS onset parameter is part of the fitting procedure. Responses to HS concentrations below 500mM were fitted simultaneously with a 500mM response from the same cell, to prevent underestimation of the RRP and overestimation of release rates. The release rate constant during spontaneous release was obtained by dividing the mEPSC frequency by the number of vesicles in the RRP. The latter was calculated by dividing the RRP charge by the average mEPSC charge. $Ca^{2+}$-dependent release rates were simulated using the previously described allosteric model (*Lou et al., 2005*). An RRP size of 15,000 vesicles was chosen based on 500mM HS responses in Syt1 WT. The rate constant $l_+$ ($2.667 \times 10^{-4}$ $s^{-1}$) was set to match spontaneous release in Syt1 WT at 0mM extracellular $Ca^{2+}$ and 20μM BAPTA-AM. All other parameters ($f$: 31.3; $k_{on}$: $1 \times 10^8$ $M^{-1}$ $s^{-1}$; $k_{off}$: 4,000 $s^{-1}$; cooperativity factor, $b$: 0.5) were set as previously described [1]. Parameters specific to Syt1 D232N ($l_{+,new}$: 0.00149; $f_{D232N}$: 62.6) were adapted from Syt1 WT parameters as described in the results section. Statistical significance was determined using Wilcoxon signed-rank tests and Mann-Whitney $U$ tests to compare paired- and independent measurements, respectively, p-values below 0.05 were considered significant. All statistical tests were performed in Matlab (Mathworks).

## Acknowledgements

We thank Desiree Schut, Lisa Laan, and Frank den Oudsten for producing glia feeders and primary culture assistance, Joke Wortel for animal breeding, Frank den Oudsten and Joost Hoetjes for genotyping, Robbert Zalm for cloning and producing viral particles, Jurjen Broeke and Hans Lodder for technical assistance.

# Additional information

## Funding

| Funder | Grant reference number | Author |
| --- | --- | --- |
| H2020 European Research Council | ERC Advanced Grant | Matthijs Verhage |
| H2020 European Research Council | 322966 | Matthijs Verhage |
| Nederlandse Organisatie voor Wetenschappelijk Onderzoek | CLS2007 | Lennart Niels Cornelisse |
| Nederlandse Organisatie voor Wetenschappelijk Onderzoek | 635100020 | Lennart Niels Cornelisse |

The funders had no role in study design, data collection and interpretation, or the decision to submit the work for publication.

## Author contributions

Vincent Huson, Data curation, Software, Formal analysis, Validation, Investigation, Visualization, Methodology, Writing - original draft, Writing - review and editing; Marieke Meijer, Rien Dekker, Data curation, Formal analysis, Investigation; Mirelle ter Veer, Marvin Ruiter, Data curation, Investigation; Jan RT van Weering, Data curation, Supervision, Methodology; Matthijs Verhage, Conceptualization, Supervision, Funding acquisition, Writing - review and editing; Lennart Niels Cornelisse, Conceptualization, Data curation, Supervision, Funding acquisition, Investigation, Methodology, Writing - original draft, Project administration, Writing - review and editing

## Author ORCIDs

Vincent Huson (iD) https://orcid.org/0000-0002-3556-1436
Rien Dekker (iD) http://orcid.org/0000-0002-6284-3279
Lennart Niels Cornelisse (iD) https://orcid.org/0000-0001-9425-2935

## Ethics

Animal experimentation: Animals were housed and bred according to institutional and Dutch governmental guidelines, and all procedures are approved by the ethical committee of the Vrije Universiteit, Amsterdam, The Netherlands (Dierexperimentencomissie (DEC) license number: FGA11-03).

## Decision letter and Author response

Decision letter https://doi.org/10.7554/eLife.55713.sa1
Author response https://doi.org/10.7554/eLife.55713.sa2

# Additional files

## Supplementary files

• Transparent reporting form

## Data availability

All data generated or analysed during this study are included in the manuscript and supporting files.

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
