## [Decision Letter]

Thank you for submitting your article "Post-tetanic potentiation lowers the energy barrier for synaptic vesicle fusion independently of Synaptotagmin-1" for consideration by *eLife*. Your article has been reviewed by Richard Aldrich as the Senior Editor, a Reviewing Editor, and three reviewers. The reviewers have opted to remain anonymous.

The reviewers have discussed the reviews with one another and the Reviewing Editor has drafted this decision to help you prepare a revised submission. In recognition of the fact that revisions may take longer than the two months we typically allow, until the research enterprise restarts in full, we will give authors as much time as they need to submit revised manuscripts.

Cornelisse and colleagues have previously reported a method to study the final step of the neurotransmitter release process, the rates of vesicle fusion by fitting a kinetic model to hypertonic solution-induced release events determined experimentally. Assuming that the fusion energy barrier is imposed by various intermediate steps of the vesicle fusion reaction, the authors have proposed that the additive effects of overcoming this energy barrier, for example, by binding of Ca to the calcium sensor Synaptotagmin-1 (Syt1), could drive the supralinear relationship between Ca and release. Following from the previous work, Huson et al. now show that Syt1 lowers the fusion energy barrier independently of its fusion clamp function, and together with slower Ca sensor(s) for asynchronous release, Syt1 also serves to reduce the fusion energy barrier during short-term plasticity (STP).

The manuscript has been reviewed by three experts, all of whom find the study to be technically well done and very interesting. However, several major concerns have been raised. We feel that all the issues can be addressed by further analysis and careful rewriting of the manuscript without the need for additional experiments. Notably the reviewers have unanimously agreed that the clarity of the paper will be significantly improved by presenting the data first, then interpreting the data within the framework of the model. Additionally, the assumptions and limitations of the model should be clearly explained in order to make the model more useful.

The individual reviewer comments that warrant consideration for the revision are as follows:

Reviewer 1:

1) There is no evidence that the kinetic parameter k2_HS that is extracted from an empirical fit to HS-induced release profile has the same Arrhenius dependence as the rate constant for evoked (AP-induced) release k2_ev (which is not even evaluated). Such evidence could be obtained by varying temperature and making Arrhenius plots for the two rate constants. It is also unclear why the frequency factor A should be fixed. It could be, e.g. calcium-dependent (and depend on calcium non-linearly). However, in light of the extraordinary situation, these experiments are not necessary, nor likely feasible in the next few months. In any case, the authors should cite some temperature studies from the literature; these would at least indicate what energy barriers are at play, and how they compare to the energy reductions the authors estimate for the various conditions (by the way, I could not find any units for energy either in the text nor figures. If we assume kJ/mole, then the reductions observed are only a few kT).

2) HS-induced release is a convenient tool to measure the amount of readily-releasable vesicles (RRP), but we run into a conceptual problem (and confusion) in Syt1 KOs (and possibly other mutants). The pool that is releasable by ~500 mM sucrose correlates well with the size of the pool releasable by AP trains in wild-type neurons. But the correlation breaks down in Syt1 KOs: the HS-induced pool is nearly the same, but AP-induced releasable pool is much smaller. This is not surprising, because HS-induced release is calcium-independent, whereas AP-induced release requires calcium (and its sensor(s)). RRP (estimated by HS application) is often taken to mean the pool of "primed" vesicles, but can we have priming without Syt1? These issues arise when we push the interpretation of HS-induced release too far. As a result, the model creates more issues than it clarifies. The simplest way to deal with this is to first present the results (some of which are interesting on their own) in a model-independent manner, and then present a possible interpretation within the framework of the model (and explaining the limitations of the model). The alternative (framing everything around the model) would require much more work to put the model on more solid ground (and may fail).

3) Evidence for a second sensor is very circumstantial. Syt7 is mentioned as a strong candidate. Does overexpression of Syt7 lead to more facilitation? If yes, this could lend support to the claim that PTP is due to two sensors that are activated simultaneously.

If Syt7 or any other calcium sensor (in addition to Syt1) is the factor "g" that binds the vesicle in Figure 1C, why does it not interact with calcium, like Syt1?

Similarly, for Syt1 D232N, how would the scheme be modified, and what calcium cooperativity would be predicted? How does the predicted cooperativity compare with the actual one (from the literature)?

4) Subsection “Synaptotagmin-1 inhibits spontaneous release without changing the fusion energy barrier”. The claim that despite its inhibitory effect on spontaneous release, Syt1 that is not calcium-bound does not increase the energy barrier for synaptic vesicle fusion, seems to contradict the basic assumption of the model (release rate ~exp(E/RT)). The problem is that the fraction of the RRP depleted by 250 mM sucrose is taken as a proxy for the energy barrier height. The curves in Figure 2K,L, cross at 250 mM sucrose, so by this criterion, the energy barrier is not changed. This is an example of over-reliance on the model to interpret the data.

The conclusion that "the spontaneous release rate can be modulated independently from the energy barrier" does not make sense. One of the basic assumptions of the model seems to be wrong (l+ being the spontaneous release rate).

Can the authors please indicate on Figure 2H (and other figures) when HS is applied? It is not clear if there is an additional delay in the KO or the traces are simply shifted.

Figure 2—figure supplement 1 top row is a repeat of Figure 2A,B,C.

5) The subtitle "Spontaneous release rates are partially regulated by energy barrier independent factors" again does not make sense. Do the authors mean spontaneous release may be due to a different pathway with its own barrier?

To avoid spontaneous Ca fluctuations leading to spontaneous fusion (a possible explanation), couldn't the authors repeat the experiments at 0 mM external calcium?

6) Discussion section: "Syt1 does not inhibit spontaneous release by increasing the fusion energy barrier" it does, at 0 mM HS. Also see the comments above.

Since Syt1's two roles on supressing spontaneous release and promoting evoked release can be dissociated with the 9Pro mutant, perhaps the authors should use the 9Pro mutant as their reference, not the WT. It would be interesting to see a replot of data vs 9Pro.

7) Discussion section: There is some vague discussion about electrostatics, membrane bending and effects of curvature that might affect the energy barrier for fusion. Is the D232N mutant able to penetrate bilayers in response to calcium binding to C2? Or is it constitutively inserted into the membrane?

8) Subsection “Fusion energy barrier model for STP”, "Fusion energy barrier model for STP"

"multiplication of membrane bound Syt1 and Syt7 corresponds to the facilitation produced by Syt7". Syt1*Syt7=facilitation?

Reviewer 2:

1) The authors investigated the relationship between evoked release and changes in the fusion energy barrier in Syt1KO cultures overexpressing the SytD232N mutant to find a convincing correlation between the increase in evoked release and the lowering of the energetic barrier. They had more difficulties to demonstrate a similar straightforward relationship with the Syt1 4W mutant, because the decrease in the energetic barrier is only detected for low HS concentrations (250mM), perhaps reflecting that the readout of the approach to measure the fusion energy barrier is more complex than initially envisioned. I think it would be helpful and informative to state those limitations which, however, do not invalidate the approach.

2) They observed higher spontaneous release in Syt1KO and Syt1 9Pro expressing synapses which is difficult to conciliate with their measurements indicating that the fusion energy barrier is unchanged. The explanation of rapid local fluctuations resistant to BAPTA in 0mM Ca^2+^ solution went too much in the limit in my opinion and it should require experimental demonstration or at least to be well documented in the literature. I wonder if such hypothetical Ca^2+^ fluctuations could be abolished using higher BAPTA concentrations. On the other hand, it could be that the changes in the fusion energy barrier in the context of spontaneous release are different (or not picked up) to the changes in the context of HS stimulation or different pool of vesicles are involved in each condition.

3) To interrogate the involvement of Syt1 in PTP makes sense in light of a previous publication by Verhage's group showing PKC-dependent phosphorylation of Syt1 in short term plasticity. The part of the study devoted to the analysis to the fusion energy barrier involvement in PTP is the most comprehensive, relevant and convincing part of the study in which conclusions are best supported by the results. It is very interesting and i opens new perspectives to investigate potential PTP mediators and its role in the modulation of the fusion energy barrier in future studies.

Reviewer 3:

1) Although the model is interesting, it is difficult to evaluate the values such as k2, max and δ E calculated from the model. It may be nice to show raw data such as time to peak, delay, integral of EPSCs and its dependence on sucrose concentration. What I mean here is that the overall results should be verified from experimental results, independent of modelling.

2) It is a bit difficult to understand why the RRP size looks smaller in Syt1 KO and Syt14W mutants but not in D232N (at least it looks like this from representative traces). The authors need some arguments. This may affect calculation of release probability.

3) In some experiments, the sucrose concentration was raised to 1M, but in some experiments, it was raised to 500 mM. Why?

4) The authors describe that spontaneous mEPSC rates are higher in Syt1 KO, but others did not see the effect, though they used the same preparation. The authors did not provide an explanation and it is puzzling.

5) Post-tetanic potentiation and the effects of phorbol esters are known to be regulated by PKC, Munc13/18, and are known to be independent of synaptotagmins. The authors need more explanation for novelty here.

6) Phorbol esters are known to change the RRP size (Stevens and colleagues), but the authors have seen somewhat different (representative traces seem to indicate the increase in the RRP size). The authors need some explanation.

7) Figure 2—figure supplement 1 top panel. The same figure as Figure 2?

---

## [Author Response]

Reviewer 1:1) There is no evidence that the kinetic parameter k2_HS that is extracted from an empirical fit to HS-induced release profile has the same Arrhenius dependence as the rate constant for evoked (AP-induced) release k2_ev (which is not even evaluated). Such evidence could be obtained by varying temperature and making Arrhenius plots for the two rate constants. It is also unclear why the frequency factor A should be fixed. It could be, e.g. calcium-dependent (and depend on calcium non-linearly). However, in light of the extraordinary situation, these experiments are not necessary, nor likely feasible in the next few months.

We agree that a direct comparison between HS-induced and AP-induced release rates at different temperatures would be interesting. However, the interpretation would be complicated since AP-induced release depends on more factors than the energy barrier alone, such as AP-shape, kinetics of the Ca^2+^ channels, Ca^2+^ binding to the sensor, and calcium dynamics, which are all temperature sensitive. We and others have previously studied the effect of temperature on AP-induced release (Huson et al., 2019; Pyott and Rosenmund, 2002) and found for instance that peak release-rates were not affected by a ten-degrees increase in temperature, while charge transfer was even reduced at higher temperatures. This was best explained by more effective Ca^2+^ buffering and Ca^2+^ clearance at higher temperatures (Huson et al., 2019). To account for all these confounding temperature effects of processes not related to the fusion barrier would require several complicated experiments that we feel are beyond the scope of this paper. We thank the reviewer for acknowledging the extraordinary situation we are currently in, which prevents executing these experiments in due time.

In any case, the authors should cite some temperature studies from the literature; these would at least indicate what energy barriers are at play, and how they compare to the energy reductions the authors estimate for the various conditions (by the way, I could not find any units for energy either in the text nor figures. If we assume kJ/mole, then the reductions observed are only a few kT).

We use RT as units for energy, which corresponds to 0.585 kcal/mol. We have now cited recent studies that have either used lipid bilayer fusion assays at different temperatures or course-grained simulations for an estimate of the fusion barrier, which was found to be around 30 RT (Discussion section). For comparison, we included values for the energy barrier reductions we found for the various conditions in the text, which all ranged between 0.2 to 0.6 RT. We discuss how these relatively small changes compare to the energy barrier effects that were predicted for the activation of Syt1 by Ca^2+^ binding (Discussion section), and how it can explain most of the increase in release probability during PTP or PDBU stimulation, in line with our model (Discussion section).

2) HS-induced release is a convenient tool to measure the amount of readily-releasable vesicles (RRP), but we run into a conceptual problem (and confusion) in Syt1 KOs (and possibly other mutants). The pool that is releasable by ~500 mM sucrose correlates well with the size of the pool releasable by AP trains in wild-type neurons. But the correlation breaks down in Syt1 KOs: the HS-induced pool is nearly the same, but AP-induced releasable pool is much smaller. This is not surprising, because HS-induced release is calcium-independent, whereas AP-induced release requires calcium (and its sensor(s)). RRP (estimated by HS application) is often taken to mean the pool of "primed" vesicles, but can we have priming without Syt1? These issues arise when we push the interpretation of HS-induced release too far. As a result, the model creates more issues than it clarifies. The simplest way to deal with this is to first present the results (some of which are interesting on their own) in a model-independent manner, and then present a possible interpretation within the framework of the model (and explaining the limitations of the model). The alternative (framing everything around the model) would require much more work to put the model on more solid ground (and may fail).

We agree that different definitions of priming exist, which may lead to different interpretations of the data. We stated more explicitly in the text (Results section) which definition we used. To circumvent unnecessary confusion, we adopted the suggestion of the reviewer to rewrite the paper and present the data first before presenting the model and interpreting the data in this framework. In addition, we explain the limitations of the model more extensively (Discussion section). We feel this has improved the clarity of the paper and balanced the discussion about the model.

3) Evidence for a second sensor is very circumstantial. Syt7 is mentioned as a strong candidate. Does overexpression of Syt7 lead to more facilitation? If yes, this could lend support to the claim that PTP is due to two sensors that are activated simultaneously.If Syt7 or any other calcium sensor (in addition to Syt1) is the factor "g" that binds the vesicle in Figure 1C, why does it not interact with calcium, like Syt1?

We briefly mention Syt7 as a sensor for paired-pulse facilitation (PPF) in the context of a more general energy barrier framework for STP. We are not aware of overexpressing studies but cite several studies that show that ablation of Syt7 did not affect normal transmission but eliminated PPF (Discussion section). However, we did not intent to suggest that Syt7 is a sensor for PTP, which occurs at a slower time scale than PPF and most likely involves a different pathway. To emphasize that the data and model we present here is for PTP we explicitly say so in the text (Results section) and in the discussion (Discussion section). Since we argue that the DAG pathway is most likely involved in the production of PTP we have adapted Figure 7C (old Figure 1C) accordingly, showing the activation of the second sensor by DAG.

Similarly, for Syt1 D232N, how would the scheme be modified, and what calcium cooperativity would be predicted? How does the predicted cooperativity compare with the actual one (from the literature)?

We have explored two mechanisms in the model by which the D232N mutant can increase Ca^2+^ sensitivity, and included a new supplemental figure (Figure 7—figure supplement 1) with a modified scheme. Based on an electrostatic energy barrier model proposed by the Sorensen lab (Ruiter et al., 2019) we modeled the reduction of the energy barrier, as a consequence of adding a charge to the C2A domain, as half of the reduction associated with the binding of one Ca^2+^ ion, which adds two charges. In addition, we modeled the two-fold increase in Ca^2+^-dependent SNARE binding found by Pang et al. (Pang et al., 2006) as a two-fold increase in factor f. These changes resulted in an increase in Ca^2+^ sensitivity, with higher release rates at all Ca^2+^ concentrations, with little effect on the apparent cooperativity (see new text in the Results section). We discuss how these predictions compare to previous findings in the literature (Pang et al., 2006; Xu et al., 2009) (Discussion section). We conclude that both mechanisms can explain the increase in Ca^2+^ sensitivity without big changes in cooperativity, but that increased Ca^2+^-dependent SNARE binding is more likely, given the other results in this study, and previous findings in literature (Discussion section).

4) Subsection “Synaptotagmin-1 inhibits spontaneous release without changing the fusion energy barrier”. The claim that despite its inhibitory effect on spontaneous release, Syt1 that is not calcium-bound does not increase the energy barrier for synaptic vesicle fusion, seems to contradict the basic assumption of the model (release rate ~exp(E/RT)). The problem is that the fraction of the RRP depleted by 250 mM sucrose is taken as a proxy for the energy barrier height. The curves in Figure 2K, L, cross at 250 mM sucrose, so by this criterion, the energy barrier is not changed. This is an example of over-reliance on the model to interpret the data.The conclusion that "the spontaneous release rate can be modulated independently from the energy barrier" does not make sense. One of the basic assumptions of the model seems to be wrong (l+ being the spontaneous release rate).

We agree that our conclusion that "the spontaneous release rate can be modulated independently from the energy barrier" is strictly speaking not correct. All fusion events occur by overcoming the fusion barrier, and increasing spontaneous release requires always that the energy barrier is reduced, at least locally and temporarily in individual synapses. What we want to argue here is that the inhibition of spontaneous release by Syt1 is not caused by an increase in the energy barrier in the presence of Syt1, as assessed with HS. We use more precise wording to describe this observation in the result section. Based on this finding we conclude indeed that the assumption in the allosteric model (Lou et al., 2005) that l+ represents the spontaneous release rate constant does not hold for hippocampal autapses, and that other mechanisms also contribute the mEPSC frequency (Discussion section). In our previous paper (Schotten et al., 2015) we came to the same conclusions based on a different data set. In our answer to point 5 below and in the Discussion section we further elaborate on possible mechanisms that contribute to spontaneous release.

Can the authors please indicate on Figure 2H (and other figures) when HS is applied? It is not clear if there is an additional delay in the KO or the traces are simply shifted.

To avoid unnecessary empty space in the figures, the traces are plotted on top of each other and indeed shifted. There is no additional delay in the KO. We have quantified the delay with respect to the onset, as well as other kinetic properties of the traces (charge integral, 10-90% rise time, time-to-peak), for all HS responses and added these data in supplemental figures to the original figures. In all plotted traces HS is applied at the start of the displayed trace for 7s (until the onset of the decay back to base-line in the traces). We tried to indicate the application interval in the graphs but felt that this did not added to the clarity of the figures. Instead we now explicitly mention in the figure captions when HS was applied.

Figure 2—figure supplement 1 top row is a repeat of Figure 2A,B,C.

These panels were left in Figure 2—figure supplement 1 by mistake and have been removed.

5) The subtitle "Spontaneous release rates are partially regulated by energy barrier independent factors" again does not make sense. Do the authors mean spontaneous release may be due to a different pathway with its own barrier?To avoid spontaneous Ca fluctuations leading to spontaneous fusion (a possible explanation), couldn't the authors repeat the experiments at 0 mM external calcium?

We agree that this statement is factually not correct (as discussed above), and changed to subsection “Changes in spontaneous release do not directly correspond to changes in the energy barrier as assessed with hypertonic sucrose”. As described in our previous paper (Schotten et al., 2015), and now more extensively explained in the Discussion section, we suggest that stochastic calcium fluctuations could contribute to spontaneous release by briefly activating Ca sensors locally in individual synapses. In this scenario, changes in the frequency of Ca fluctuations or in the sensitivity of the release machinery to these fluctuations, e.g. by unclamping a high-affinity Ca sensor, would also change the spontaneous fusion rate, but in an additive manner, not in a multiplicative manner as l+ would do. These additive changes in release rate would be negligible compared to release rates induced by sucrose and not translate to changes in the fusion barrier.

Our sucrose experiments with the 9Pro mutant were performed in 0 mM external calcium and in the presence of 20µM BAPTA-AM, to avoid spontaneous Ca fluctuations, as also suggested by the reviewer. We discuss these results and possible interpretations now in more detail in the Discussion section. Under these conditions spontaneous release was reduced in WT glutamatergic autapses, but to a lesser extent than reported for GABAergic neurons it seemed (Xu et al., 2009), although we did not do a direct comparison. Also, spontaneous release remained significantly higher in the 9Pro mutant compared to WT, under these conditions. This might suggest that BAPTA loading was insufficient to block all spontaneous events. We also include an alternative explanation, as suggested by this reviewer and reviewer 2, that spontaneous release occurs through a separate pathway. This opens the possibility that effects on the fusion barrier of genetic manipulations that increase spontaneous release could be missed with our sucrose assay when only a small subset of vesicles is affected, or when these vesicles are somehow less sensitive to our HS assay (Discussion section).

6) Discussion section: "Syt1 does not inhibit spontaneous release by increasing the fusion energy barrier" it does, at 0 mM HS. Also see the comments above.

This relates to point 4 and 5 discussed above. In agreement with our answers to these points we have adapted the title to "Syt1 does not inhibit spontaneous release by increasing the fusion energy barrier as assessed with hypertonic sucrose".

Since Syt1's two roles on supressing spontaneous release and promoting evoked release can be dissociated with the 9Pro mutant, perhaps the authors should use the 9Pro mutant as their reference, not the WT. It would be interesting to see a replot of data vs 9Pro.

We agree that the 9Pro mutant is of important value to study the inhibitory role of Syt1 separately from its activating role. This is why we included it in our study to investigate the relation between changes in spontaneous release and changes in the fusion barrier. For research questions concerning the activation role we think the unmutated Syt1 WT is a better control. Moreover, we always make sure for all our experiments that cell cultures we use for experimental and control groups are from the same preparation (KO-rescue experiments) or littermate pups from the same week (KO vs WT experiments). Therefore, we cannot replot our data using the 9Pro data from different experiments as a control, and many new experiments would be required, which is not feasible given the current COVID situation.

7) Discussion section: There is some vague discussion about electrostatics, membrane bending and effects of curvature that might affect the energy barrier for fusion. Is the D232N mutant able to penetrate bilayers in response to calcium binding to C2? Or is it constitutively inserted into the membrane?

Biochemical properties of the Syt1 D232N mutant were studied in (Pang et al., 2006). The ability to penetrate bilayers, either in response to Ca or constitutively, was not directly tested. However, no difference was found in phospholipid binding at different Ca-concentrations between Syt1 WT and D232N. In hindsight, this is not in line with one of our suggestions in the discussion that increased membrane interactions of this mutant would reduce the fusion barrier by expelling interlamellar water, which was therefore left out. This also clarifies the discussion.

8) Subsection “Fusion energy barrier model for STP”, "Fusion energy barrier model for STP""multiplication of membrane bound Syt1 and Syt7 corresponds to the facilitation produced by Syt7". Syt1*Syt7=facilitation?

This refers to the observation in (Jackman and Regehr, 2017) that the amount membrane bound Syt1 multiplied with the amount of membrane bound Syt7, when taken as a proxy for sensor activation, can account for the facilitation observed in the same synapses. This is in line with the energy barrier model for STP, which predicts that in a dual sensor model additive effects of different sensors on the fusion barrier lead to multiplicative effects on the fusion rate. We have clarified this in the Discussion section.

Reviewer 2:1) The authors investigated the relationship between evoked release and changes in the fusion energy barrier in Syt1KO cultures overexpressing the SytD232N mutant to find a convincing correlation between the increase in evoked release and the lowering of the energetic barrier. They had more difficulties to demonstrate a similar straightforward relationship with the Syt1 4W mutant, because the decrease in the energetic barrier is only detected for low HS concentrations (250mM), perhaps reflecting that the readout of the approach to measure the fusion energy barrier is more complex than initially envisioned. I think it would be helpful and informative to state those limitations which, however, do not invalidate the approach.

This is a valid point. We have addressed the limitations of our HS method now more explicitly in the discussion. We restructured our discussion on the possible interaction between the mechanisms by which HS and Syt1 reduce the fusion energy barrier, as an explanation for why changes in the energy barrier are not detected beyond 250mM for the Syt1 mutants (in contrast to energy barrier changes after PDBU or PTP, which are detected at all sucrose concentrations). We also address more specifically why the enhanced hydrophobicity of Syt1 4W could have opposite effects at higher sucrose concentrations (Discussion section). Furthermore, also in response to point 5 of reviewer 1 and point 2 of this reviewer, we discuss how our method is insensitive to possible differential effects on evoked and spontaneous release if these are governed by different pathways/vesicle pools (Discussion section).

2) They observed higher spontaneous release in Syt1KO and Syt1 9Pro expressing synapses which is difficult to conciliate with their measurements indicating that the fusion energy barrier is unchanged. The explanation of rapid local fluctuations resistant to BAPTA in 0mM Ca^2+^ solution went too much in the limit in my opinion and it should require experimental demonstration or at least to be well documented in the literature. I wonder if such hypothetical Ca^2+^ fluctuations could be abolished using higher BAPTA concentrations. On the other hand, it could be that the changes in the fusion energy barrier in the context of spontaneous release are different (or not picked up) to the changes in the context of HS stimulation or different pool of vesicles are involved in each condition.

This is a fair point, similar to point 5 of reviewer 1. Since other studies were successful in reducing spontaneous release by 95% using 10uM BAPTA-AM and 0mM Ca, we considered our explanation of local Ca fluctuations being BAPTA resistant less likely and took this out of the discussion. We now consider the possibility that our BAPTA loading was not sufficient to block all spontaneous events (Discussion section). We also included an alternative explanation, as suggested by this reviewer and reviewer 1, that spontaneous release occurs through a separate pathway. This opens the possibility that effects on the fusion barrier of genetic manipulations that increase spontaneous release could be missed with our sucrose assay when only a small subset of vesicles is affected, or when these vesicles are somehow less sensitive to our HS assay (Discussion section).

3) To interrogate the involvement of Syt1 in PTP makes sense in light of a previous publication by Verhage's group showing PKC-dependent phosphorylation of Syt1 in short term plasticity. The part of the study devoted to the analysis to the fusion energy barrier involvement in PTP is the most comprehensive, relevant and convincing part of the study in which conclusions are best supported by the results. It is very interesting and i opens new perspectives to investigate potential PTP mediators and its role in the modulation of the fusion energy barrier in future studies.

We thank the reviewer for these kind remarks and agree that our findings could shed new light on mechanisms that drive STP.

Reviewer 3:1) Although the model is interesting, it is difficult to evaluate the values such as k2, max and δ E calculated from the model. It may be nice to show raw data such as time to peak, delay, integral of EPSCs and its dependence on sucrose concentration. What I mean here is that the overall results should be verified from experimental results, independent of modelling.

We think we might have created some confusion due to the original layout of the paper, where we first presented the dual sensor model and then the data. The dual sensor model we present in this paper is not the model we used to fit HS responses. Instead we used a simpler release model, without sensors, in our fitting method that we previously developed (Schotten et al., 2015). To avoid this confusion, and also in response to point 2 of reviewer 1, we changed the order of the manuscript, and first present the data before presenting the dual sensor model. In our previous paper (Schotten et al., 2015), to which this study is follow up, we have put a lot of effort to optimize and validate our fitting method, which we feel has two advantages over other analysis methods. Firstly, it quantifies parameters such as RRP size, priming rate constant, fusion rate constant and energy barrier changes, which give direct information about (changes in) the release process. Secondly, it allows for more accurate quantification of release parameters: RRP is determined more precisely, due to a more realistic correction for vesicle replenishment, and arbitrary choices about the onset of the HS response, introducing additional variation in parameters such as “time-to-peak” and “delay” are circumvented. We have emphasized these advantages in the Materials and methods section. For these reasons, we choose our fit method over other analysis methods, and felt that a presentation of more HS parameters would complicate the text further. Nevertheless, for completeness and comparison with other HS studies we included seven new supplementary figures, where we provide statistics of the HS integral (integral of the HS EPSC), 10-90% rise-time, delay of HS onset (delay of the onset of the HS induced current with respect the start of the HS stimulus), time-to-peak (time of the peak current with respect to the start of the HS stimulation), and time-to-peak minus delay.

2) It is a bit difficult to understand why the RRP size looks smaller in Syt1 KO and Syt14W mutants but not in D232N (at least it looks like this from representative traces). The authors need some arguments. This may affect calculation of release probability.

We did not find significant changes in RRP size for any of these mutants (Figure 1I, Figure 3—figure supplement 2G). The HS responses for 500mM and higher might seem smaller in the Syt1 KO and Syt1 4W, given the lower peaks in the typical examples, but these are the consequence of slower release rates at these concentrations (Figure 1K, Figure 3—figure supplement 2I). Slower release kinetics produce lower, but wider peaks, leaving the “area under the curve”, used for fitting of the RRP (Schotten et al., 2015), the same. Interestingly, release kinetics in the D232N mutant is slightly higher at 500mM, producing higher and sharper peaks. We tried to find the most representative traces for our typical examples, but it has proven difficult to always exactly match all features in the same example. This is due to the variation in the data, as indicated by the individual data points in the boxplots, and the fact that we always select our example traces for all HS concentrations from the same cell.

3) In some experiments, the sucrose concentration was raised to 1M, but in some experiments, it was raised to 500 mM. Why?

The two experiments the reviewer is referring to were carried out at the beginning of this study and covered a broader range of sucrose concentrations to include data points beyond 500mM, since we did not know exactly what to expect. During the course of the study we found that stimulations of 750mM and 1M were more challenging than 500mM and less, while the assay was most sensitive at concentrations around 250mM. Therefore, to be efficient, we carried out most of our experiments with 250mM and 500mM sucrose during the rest of the study.

4) The authors describe that spontaneous mEPSC rates are higher in Syt1 KO, but others did not see the effect, though they used the same preparation. The authors did not provide an explanation and it is puzzling.

Our finding that spontaneous release is increased autapses in the absence of Syt1 was recently confirmed by an independent study form our institute (Diez-Arazola et al., 2020), and a study from the Sorensen lab (Ruiter et al., 2019), and is in line with previous studies in network cultures. It remains enigmatic why the latter group and the Chapman lab previously found a discrepancy between the Syt1 KO phenotype in networks and autapses. It is difficult for us to speculate about a specific explanation without information about the experimental details in these studies. In the discussion we have included a reference to the new autapse studies and provide two general explanations for the difference in Syt1KO autapse phenotypes (Discussion section).

5) Post-tetanic potentiation and the effects of phorbol esters are known to be regulated by PKC, Munc13/18, and are known to be independent of synaptotagmins. The authors need more explanation for novelty here.

Although, the involvement of PKC, Munc13, and Munc18 in PTP and phorbol ester potentiation has been studied, it was not known whether Syt1 is required for the reduction in the energy barrier after PTP and phorbol ester stimulation. Our experiments are the first to show that modulation of the fusion barrier by PTP/phorbol esters can occur in the absence of Syt1, which is an important assumption of our energy barrier model for PTP. We now briefly explain the rationale of these experiments in the text (Results section).

6) Phorbol esters are known to change the RRP size (Stevens and colleagues), but the authors have seen somewhat different (representative traces seem to indicate the increase in the RRP size). The authors need some explanation.

Stevens and Sullivan, 1998 indeed reported increased RRP size after PDBu application, measured with 500mM sucrose. However, several later studies did not replicate this finding and reported unchanged pool sizes after phorbol ester application (Basu et al., 2007; Lou et al., 2008; Schotten et al., 2015; Wierda et al., 2007). We mention this now in the Discussion section. Responses to 500mM sucrose in the typical examples might appear larger after PDBU application due to faster release kinetics of release. This produces sharper and higher peaks, while the total charge transfer is not changed. Furthermore, results for individual cells showed some variation in their response to PDBU as depicted by the individual data points in Figure 6E. Since we always select our example traces for all conditions (250mM, 500mM, +/- PDBU) from the same cell, some examples might deviate slightly from the average values.

7) Figure 2—figure supplement1 top panel. The same figure as Figure 2?

These panels were left in Figure 2—figure supplement 1 by mistake and have been removed.